# RUBCN as a novel prognostic biomarker and therapeutic target in breast cancer

**Dong Dong Yang**[1,©], **Cheng Hao Liu**[1,©], **Sheng Qiu Jia**[1], **Ze Kuan Xue**[1], **Ming Ming Zhang**[2], **Rui Yang**[3], **Yong Zhou Huang**[4], **Xin Chun Zhao**[4], **Bao San Han**[5,6], **Sheng Dong Nie**[7], **Gui Lin Huang**[1,4*], **Ji Xue Hou**[1,4*]

1 School of Medicine, Shihezi University, Shihezi, China, 2 Department of Internal Medicine, Affiliated Cancer Hospital of Zhengzhou University, Henan Cancer Hospital, Zhengzhou, China, 3 Department of Breast and Thyroid Surgery, Union Hospital, Tongji Medical College, Huazhong University of Science and Technology, Wuhan, China, 4 Department of Thyroid and Breast Surgery, The First Affiliated Hospital of Shihezi University, Shihezi, China, 5 Department of Breast Surgery, Xinhua Hospital Affiliated of Shanghai Jiao Tong University School of Medicine, Shanghai, China, 6 Research Center of Breast Tumor Intelligent Diagnosis and Treatment, University of Shanghai for Science and Technology, Shanghai, China, 7 University of Shanghai for Science and Technology, School of Health Science and Engineering, Shanghai, China

© These authors contributed equally to this work.
* 15559016883@163.com (GLH); hjx1506@163.com (JXH)

## Abstract

Analysis of autophagy-related gene expression data identified RUBCN as a novel biomarker influencing the pathogenesis and progression of breast cancer, underscoring its potential as a therapeutic target. We analyzed multiple breast cancer sample datasets using bioinformatics tools and databases. A consensus prognostic model was constructed and validated across several independent datasets to further examine its association with patient outcomes. A series of bioinformatics analyses focused on RUBCN were conducted, including expression profiling, independent prognostic evaluation, immune correlation analysis, and survival analysis. RUBCN expression was verified in breast cancer cell lines and clinical tissue specimens via Western blotting, quantitative real-time reverse transcription PCR, and immunohistochemistry. Functional assays, such as the Cell Counting Kit-8 assay, 5-ethynyl-2'-deoxyuridine incorporation assay, wound healing assay, and Transwell invasion assay, were employed to evaluate the effects of RUBCN knockdown on breast cancer cell proliferation and invasion. Autophagic activity, indicated by LC3 and P62 levels, was measured via Western blot in RUBCN-knockdown breast cancer cells with or without chloroquine treatment. Elevated expression of multiple autophagy-related genes was observed in breast cancer. The consensus prognostic model accurately predicted survival across multiple datasets, with RUBCN emerging as a key gene whose expression levels were significantly correlated with patient prognosis. Enrichment analysis indicated that RUBCN likely promotes breast cancer progression by regulating cell cycle and invasion processes. Further investigation revealed a negative

**Data availability statement:** All data files are available from the TCGA,GEO and HGNC database (accession number(s) GSE9893 (GPL5049, n=155), GSE20685 (GPL570, n=327), GSE20711 (GPL570, n=90), GSE22219 (GPL6098, n=216), GSE25065 (GPL96, n=198), GSE61304 (GPL570, n=62), and GSE162228 (GPL570, n=133).).Publicly available datasets were analyzed in this study. This data can be found here: https://www.genenames.org/. The custom code used to analyze the data is available in the following GitHub repository: https://github.com/ydd660/fromto.git.

**Funding:** This work was supported by the National Natural Science Foundation of China (Grant No. 82260105), Project supported by the Science and Technology Program of Xinjiang Production & Construction Corps, China (Grant No. 2024AB065) and Hospital-Level Research Project (Grant No. BS202202). The funders had no role in the study design, data collection and analysis, decision to publish, or preparation of the manuscript.

**Competing interests:** The authors have declared that no competing interests exist.

correlation between RUBCN expression levels and immune cell infiltration, suggesting a potential role in mammary tumorigenesis through mediating immune evasion by suppressing immune cell infiltration. Immunohistochemical results confirmed upregulated RUBCN expression in carcinoma tissues. Knockdown of RUBCN was shown to suppress the proliferative and invasive abilities of breast cancer cells. Mechanistically, RUBCN knockdown impaired autophagic flux, as evidenced by altered LC3 and P62 levels upon chloroquine treatment. Together, these findings establish RUBCN as a promising therapeutic target in breast cancer. Future studies should emphasize in vivo functional validation using animal models and screen for targeted agents capable of modulating RUBCN expression or activity, thereby facilitating the development of innovative therapeutic strategies for breast cancer treatment.

## Introduction

Since the late 1970s, the global incidence of breast cancer has exhibited a sustained upward trend. According to World Health Organization (WHO) statistics, by 2020, breast cancer had surpassed lung cancer to emerge as the most prevalent malignancy among women worldwide [1]. Currently, breast cancer classification primarily employs two major methodologies: histopathological typing and molecular subtyping. Histopathological classification categorizes breast cancer into noninvasive carcinoma, early invasive carcinoma, and invasive carcinoma based on the extent of tumor cell invasion into surrounding tissues and metastatic potential. Molecular subtyping defines four principal categories: Luminal A, Luminal B, HER2-overexpressing, and triple-negative breast cancer (TNBC; basal-like) [2,3]. Breast cancer exhibits high heterogeneity. Despite the availability of diverse therapeutic modalities, including surgical resection, endocrine therapy, radiotherapy, and molecularly targeted agents [4], tumor drug resistance and post-treatment recurrence remain major clinical challenges. This therapeutic dilemma highlights the urgent need for in-depth research into the pathogenesis of breast cancer. Thereby, identifying novel therapeutic targets and regulatory pathways will lay the groundwork for developing more effective therapeutic strategies [5].

Autophagy is well-established to be crucial for maintaining organismal homeostasis [6–8], with growing attention on its dual regulatory role in tumorigenesis and progression. Although autophagy functions as a tumor suppressor in the early stages of tumorigenesis by clearing dysfunctional organelles to preserve genomic integrity, it exhibits pro-tumorigenic properties in established tumors by supporting tumor cell survival under stress conditions, including nutrient deprivation or chemotherapy, through the maintenance of cellular metabolism and energy homeostasis [9]. This functional duality is regulated by a network of autophagy-related genes, which play a significant role in the pathogenesis of breast cancer [10]. Evidence for this comes from multiple fronts: for instance, the autophagy regulator ATG4A is not only linked to poor survival and therapy resistance but also crucial for maintaining cancer stem cell properties that drive tumorigenicity and metastasis [11,12]. Moreover, even

dysfunctional autophagy can be pathogenic, as cytosolic accumulation of the selective receptor NBR1 drives breast cancer metastasis [13]. Collectively, these findings underscore the complexity and criticality of the autophagy-related gene network in breast cancer. Beyond well-studied genes like ATG4A, other less-characterized autophagy regulators, such as RUBCN, may also play critical roles in breast cancer pathogenesis, warranting further investigation.

RUBCN (Rubicon), located at chromosome 3q29, comprises 20 exons. This gene encodes a ubiquitously expressed 972-amino acid protein that functions as one of the few endogenous inhibitors of macroautophagy. Originally characterized as a component of the class III PI3K complex, this multifunctional protein primarily negatively regulates macroautophagy and endolysosomal trafficking [14,15].Therefore, this study employed bioinformatics approaches to analyze and compare the expression levels of RUBCN in breast cancer tissues, investigating its correlation with patient prognosis and clinicopathological characteristics. Through GSEA and other methodologies, the potential mechanisms underlying the association between RUBCN expression and breast cancer progression were explored, aiming to provide novel theoretical insights for the treatment of breast cancer.

## Materials and methods

### Obtaining patients and acquisition of related genes

The autophagy-related gene set was curated from the official database of the HUGO Gene Nomenclature Committee (HGNC) (https://www.genenames.org) [16]. RNA sequencing (RNA-seq) data for breast invasive carcinoma (BRCA) were obtained from The Cancer Genome Atlas (TCGA) portal (https://portal.gdc.cancer.gov), processed, and transformed into transcripts per million (TPM)-normalized expression matrices. Differential expression of autophagy-related genes was quantitatively compared between tumor tissues and histologically normal mammary tissues using paired TCGA-BRCA samples. To validate survival outcomes, gene expression profiles and clinicopathological data from seven Gene Expression Omnibus (GEO) datasets were integrated: GSE9893 (GPL5049, n = 155), GSE20685 (GPL570, n = 327), GSE20711 (GPL570, n = 90), GSE22219 (GPL6098, n = 216), GSE25065 (GPL96, n = 198), GSE61304 (GPL570, n = 62), and GSE162228 (GPL570, n = 133).

### Construction and interaction pattern analysis of protein-protein interaction (PPI) networks

The protein-protein interaction (PPI) network was constructed based on the Search Tool for the Retrieval of Interacting Genes (STRING) database (https://cn.string-db.org/) [17], an online analytical platform enabling visualization and quantitative profiling of interaction relationships among differentially expressed genes (DEGs). DEGs identified in this study were imported into the Search Tool for the Retrieval of Interacting Genes (STRING) database for protein-protein interaction (PPI) network analysis. This enabled the mapping of autophagy-related protein interactions and identification of pivotal hub proteins within the network for subsequent investigations. All annotated genes were then downloaded and imported into Cytoscape (V3.8.2) software to perform module clustering analysis on the PPI network.

### Data cleaning and preprocessing

Data cleaning involved removal of missing values and exclusion of non-tumor samples to ensure data integrity and accuracy. Survival time was converted from days to years for temporal unit standardization. All validation datasets underwent z-score normalization to transform variables into a distribution with zero mean and unit variance. Subsequently, exponential transformation was applied to convert potentially negative z-scores into strictly positive values, thereby preserving non-negativity and enhancing interpretability in risk assessment models.

### Modeling algorithm and parameter optimization

Multiple survival analysis algorithms were employed for modeling, including Lasso regression (Least Absolute Shrinkage and Selection Operator), Elastic Net, Ridge regression, stepwise Cox regression, and CoxBoost [18]. Lasso regression

was implemented via the glmnet package with the family parameter specified as "cox" and alpha set to 1. Ten-fold cross-validation using the cv.glmnet function was performed to identify the optimal λ value. Model coefficients corresponding to the optimal λ and their associated feature names were extracted, followed by screening for non-zero coefficients and their corresponding gene symbols.Elastic Net and Ridge regression were similarly executed using the glmnet package. For Elastic Net, the alpha parameter ranged between 0.1 and 0.9 (indicating a blend of L1/L2 regularization) [19], while Ridge regression required alpha fixed at 0 (pure L2 regularization). Stepwise Cox regression involved initial construction of a multivariate Cox model via the coxph function, followed by variable selection using the stepAIC function with direction parameters set to "both", "forward", or "backward". For the Elastic_net_0.1 model, the glmnet package was employed to implement Elastic Net regression with the L1/L2 mixing ratio parameter α fixed at 0.1. Ten-fold cross-validation (cv.glmnet) determined the optimal regularization strength λ, and features with non-zero coefficients were subsequently extracted as signature genes.

## Evaluation of prognostic model prediction performance

Based on the Elastic Net model-derived risk scores, patients were dichotomized into high- and low-risk groups using the optimal cutoff determined by the surv_cutpointfunction (survminerpackage), with a minimum group size of 30%. The area under the receiver operating characteristic curve (ROC-AUC) served as the primary evaluation metric [20]. Time-dependent AUC values at 1, 3, and 5 years were calculated using the timeROCpackage to assess the performance of various prognostic models. To evaluate the prognostic value of the risk score, univariate Cox analysis was performed via the coxphfunction to estimate hazard ratios (HR) within each distinct dataset. Subsequently, a meta-analysis was conducted to integrate these HRs across datasets using the inverse-variance method. Finally, Kaplan-Meier curves were generated with the survfitfunction (survivalpackage), and the survival differences between the high- and low-risk groups were compared using the log-rank test to validate the prognostic stratification efficacy.

## Evaluation of RUBCN as a critical risk gene in BRCA-related pathways

Coefficients for each gene were computed using multiple algorithms and visualized in a heatmap, wherein a coefficient of zero indicated the gene's exclusion from a specific algorithm's model. Subsequently, univariate Cox survival analysis was performed for RUBCN, followed by meta-analysis employing the inverse-variance method, with the logarithm of hazard ratio (HR) as the primary effect measure. An HR < 1 indicates tumor-suppressive effects, while HR > 1 suggests oncogenic properties. Statistical analysis and visualization were implemented using the "Meta" package within the R environment.

## Correlation analysis of clinical factors and construction of a prognostic nomogram

A nomogram was constructed based on the Cox proportional hazards model using the rmspackage in R (v4.4.1), providing an intuitive visualization of the contribution of each predictor to survival risk. Parameters were customized to quantify individual predictor weights, and risk scores were calculated for each patient.

## Kaplan-Meier survival curve analysis

The optimal cut-off value for RUBCN expression was determined using the surv_cutpoint function from the "survminer" package, categorizing samples into high- and low-RUBCN expression groups. Subsequently, log-rank tests were performed via the "survival" package with a significance threshold of α = 0.05. Results were visualized using the "survminer" and "ggplot2" packages.

## GO, KEGG and GSEA

To gain a comprehensive understanding of the biological functions and pathways enriched by RUBCN, Gene Ontology (GO) and Kyoto Encyclopedia of Genes and Genomes (KEGG) [20,21] pathway enrichment analyses were performed

using the "clusterProfiler" R package (version 3.14.3). Gene Set Enrichment Analysis (GSEA) [22] is a gene set-based analytical method that evaluates the enrichment of a predefined gene set within a phenotype-correlated ranked gene list, thereby determining whether the gene set is significantly associated with a specific phenotype or biological process. Samples were stratified into high- and low-expression groups based on RUBCN expression levels, and differential expression analysis was conducted using the limma package. The logarithmic fold change (log2 Fold Change, log2FC) or Delta value of the Z-score was calculated for each gene. Following differential expression analysis, gene set enrichment analysis was performed using the GSEA function in the clusterProfiler package.

## Gene set variation analysis (GSVA)

Gene Set Variation Analysis (GSVA) [23] was performed to quantify pathway activity variations without pre-specifying sample groups. Using the CancerSEA database (http://biocc.hrbmu.edu.cn/CancerSEA/) [24], we applied the GSVA R package to compute combined z-scores for six functional state gene sets based on the gene expression matrix of dataset GSE9893. The resulting enrichment scores were standardized using the scale function to generate gene set activity scores. Pearson correlation coefficients were then calculated between individual genes and each gene set activity score to evaluate their associations.

## Spatial transcriptomic profiling of the breast cancer TME

Spatial transcriptomics was performed on two breast cancer specimens using the SpatialTME database (https://www.spatialtme.yelab.site/) [25], which employs the cotrazm package for deconvolution of tumor microenvironment (TME) cellular composition [26]. Spatial spots were annotated based on the predominant cell type within each microregion. RUBCN expression patterns across distinct cellular microregions were visualized via heatmap. Further analyses were conducted on tissue sections from two representative tumor samples. Spearman correlation analysis quantified associations between RUBCN expression and proportions of specific cell types within microregions, while differential RUBCN expression in tumor versus normal groups was assessed using Wilcoxon rank-sum test.

## Pathological sample collection and immunohistochemistry

This study included primary tumor tissues and paired adjacent normal tissues from five patients with BRCA. All samples were obtained from patients who underwent surgery at the Department of Thyroid and Breast Surgery, The First Affiliated Hospital of Shihezi University, between June 2025 and July 2025. The study was approved by the Medical Ethics Committee of The First Affiliated Hospital of Shihezi University (Approval No.: KJ2023-169-03). All adult participants provided written informed consent authorizing the use of their anonymized samples for subsequent research; no minors were involved in the study. All samples were processed using independent coding to replace patient identifiers, ensuring that personal privacy could not be traced.Surgically resected tissues were fixed in 10% neutral buffered formalin for 24–48 hours and then embedded in paraffin. Sections (4-μm thick) were baked at 60°C for 1 hour, dewaxed in xylene, and rehydrated through a graded ethanol series. For antigen retrieval, the sections were heated in citrate buffer (pH 6.0) using a microwave oven for 15 minutes and maintained at a sub-boiling temperature. After cooling, endogenous peroxidases were blocked with 3% hydrogen peroxide for 10 minutes, and nonspecific binding was blocked with 5% bovine serum albumin in PBS for 30 minutes. The sections were then incubated overnight at 4°C with the primary antibody against RUBCN (Cat. No. 21444–1-AP, Proteintech Group, Inc.; diluted at 1:50). After washing with PBS, the sections were incubated with an appropriate horseradish peroxidase (HRP)-conjugated secondary antibody for 1 hour at room temperature. The signal was visualized using DAB, and the nuclei were counterstained with hematoxylin. Finally, the sections were mounted with neutral gum. The expression level of RUBCN was evaluated using the H-score method by two independent pathologists blinded to the clinical data. The immunohistochemical staining was assessed using a semi-quantitative method. The staining intensity was categorized on a scale of 0–3: 0 (negative), 1 (weak), 2 (moderate), and 3 (strong). The extent of

staining was assessed on a scale of 0–5, based on the percentage of immunoreactive cells: 0 (0%), 1 (≤1%), 2 (2–10%), 3 (11–33%), 4 (34–66%), and 5 (≥67%). A combined score, ranging from 0 to 8, was obtained by adding the intensity and extent scores.

## Cells and reagents

Human non-transformed mammary epithelial cell line MCF-10A and breast cancer cell lines, including MCF-7, MDA-MB-468, SK-BR-3, MDA-MB-231, in this study were purchased from National Collection of Authenticated Cell Cultures (China). MCF-7 was cultured in MEM medium (containing 10% fetal bovine serum, 1% penicillin streptomycin and 0.01 mg/ml insulin). MCF-10 A was cultured with the following medium: DMEM +5% HS + 20 ng/mL EGF + 0.5 µg/mL Hydrocortisone+10 µg/mL Insulin +1% NEAA +1% P/S. SK-BR-3 was cultured using McCoy's 5A + 10% FBS + 1% P/S. MDA-MB-468 and MDA-MB-231 were cultured using DMEM medium (containing 10% fetal bovine serum and 1% penicillin streptomycin). All cell lines were cultured under 37°C, 5% CO2 condition.

## Quantitative real-time polymerase chain reaction (qRT-PCR)

The cultured cells were subjected to total RNA extraction using the E.Z.N.A.® Total RNA Kit I (Cat. No. R6834-01, OMEGA). Total RNA was reverse-transcribed into cDNA with the RevertAid First Strand cDNA Synthesis Kit (Thermo Fisher Scientific, Cat. No. K1622). Following reverse transcription, samples were loaded according to the reaction system specified by the qRT-PCR kit, with at least three technical replicates prepared per group for instrument analysis. β-actin was designated as the reference gene. The relative expression level of RUBCN in mammary cells was determined using the $2-\Delta\Delta Ct$ method. The primers used for qRT-PCR are listed in S1 Table.

## siRNA transfection

MDA-MB-231 cells were seeded in a 6-well plate at a density of $2 \times 10^5$ cells per well and transfected with siRNA using Lipofectamine 2000 (Invitrogen, Cat. No. 11668019) according to the manufacturer's protocol. After 6 h of transfection, the medium was replaced with fresh complete medium, and the cells were further cultured for 24 h. Subsequently, cellular RNA or protein was extracted for downstream functional assays. The siRNA sequences used for cell transfection experiments are listed in S2 Table.

## CCK-8 assay

Cell proliferation was assessed by CCK-8 (APExBIO, Cat. No. K1018) determination. MDA-MB-231 cells were seeded in 96-well plates at a density of $2 \times 10^3$ cells per well and cultured in an incubator for 0, 24, 48, and 72 hours. Subsequently, 10 µL of CCK-8 reagent was added to each well and incubated for 2 hours.A Multiskan FC Microplate Photometer (Thermo Fisher Scientific) was utilized to quantify the absorbance at 450 nm.

## Western blotting

Cells were lysed on ice using RIPA lysis buffer (Solarbio, Cat. No. R0010). The resulting protein lysates were separated by sodium dodecyl sulfate-polyacrylamide gel electrophoresis (SDS-PAGE) and transferred onto 0.45 µm polyvinylidene fluoride (PVDF) membranes (Millipore, Cat. No. 88518). After blocking, the membranes were incubated overnight at 4°C with primary antibodies. Antibodies used were as follows: anti-RUBCN (1:1000), anti-LC3 (1:500, Abmart, China, Cat. No. T55992), and anti-P62 (1:500, Servicebio, Cat. No. GB11239-1-50). Subsequently, the membranes were incubated with a horseradish peroxidase (HRP)-conjugated secondary antibody for 1 hour at room temperature. Protein bands were visualized using an enhanced chemiluminescence (ECL) reagent (Biosharp, Cat. No. BL520B). An anti-GAPDH antibody (1:2500, Cell Signaling Technology, CST, Cat. No. 3700) was used as a loading control. For the autophagic flux assay,

cells were treated with or without chloroquine (MCE, Cat. No. HY-17589A) prior to lysis. All Western blot analyses were performed with three independent biological replicates (n = 3).

## Colony formation assay

A cell suspension containing 1,000 cells was seeded in a 6-well plate (Corning, Cat. No. CLS3516) and cultured for 2 weeks until colonies became evident. The cells were fixed with 4% paraformaldehyde (Servicebio, G1101-500ML) for 20 minutes, stained with 1% crystal violet (Solarbio, Cat. No. G1062), and counted.

## Cell migration and invasion assays

Cells were evenly seeded into 6-well plates. After cell adhesion, a linear wound was created by scraping the monolayer with a 200 μl pipette tip. The wells were washed with PBS to remove detached cells, and images were captured at 0 and 48 hours. For the Transwell invasion assay, polycarbonate membranes (8 μm pore size; Corning, Cat. No. PTEP24H48) were used. Matrigel (Corning, Cat. No. 354234) was diluted 1:8 in complete medium and polymerized in a 37°C incubator for 3 hours. Cells were resuspended in serum-free medium, and 200 μl of the cell suspension was seeded into the upper chamber. Complete medium was added to the lower chamber. After 24 hours, cells that had migrated through the membrane were fixed with 4% paraformaldehyde and stained with crystal violet.

## EdU staining

Cells were plated uniformly in a 12-well plate and cultured for 12 hours. The culture medium was then replaced with fresh medium containing 10 μM EdU, and the cells were incubated for another 3 hours. Following fixation, subsequent procedures were performed according to the manufacturer's protocol using the EdU Imaging Kit (Cat. No. K1075) from APExBIO Technology. Stained cells were imaged under a fluorescence microscope(Olympus,10x). The percentage of EdU-positive cells was calculated by counting EdU-positive (Cy3, red) nuclei and normalizing to the total number of nuclei stained with Hoechst 33342 (blue) from at least three random fields per well.

## Statistical analysis

Statistical analyses were performed using R (v4.4.1) and GraphPad Prism (v9.5.1). In vitroexperimental results are presented as mean ± SD (n = 3). Methods for determining statistical significance are detailed in respective figure legends. Differences in survival rates among BRCA patients were analyzed using Kaplan-Meier curves with log-rank tests. A $P$-value < 0.05 was considered statistically significant.

## Results

### Differential expression and genetic variation of DEGs in BRCA

A total of 46 autophagy-related genes were obtained from the HGNC database. In the TCGA-BRCA cohort, 35 DEGs were identified when compared to normal breast tissues, including 21 upregulated and 14 downregulated genes in tumor samples. The following genes demonstrated increased expression levels in breast tumors (Fig 1A): ATG3, ATG7, ATG9B, ATG101, RUBCN, CERKL, AMBRA1, ATG4A, ATG4B, ATG4D, ATG12P1, ATG12P2, ATG16L1, ATG16L2, ATG4AP1, ULK1, BECN2, VMA21, CISD2, VMP1, and C16orf70. In contrast, the following genes exhibited significantly reduced expression in breast cancer tissues (Fig 1B): ATG2B, ATG9A, ATG10, ATG14, DEPP1, NBR1, RUBCNL, ATG4C, DRAM1, DRAM2, EPG5, FYCO1, ULK2, and ELAPOR2.Somatic mutations were detected in 704 of 1, 103 samples (64%). Among these, BECN2, VMP1, and RUBCN exhibited the highest mutation frequencies (Fig 1C). The PPI network was constructed to delineate the complex interplay among DEGs, revealing statistically significant correlations in 25 out of 29 analyzed genes (Fig 1D). The chromosomal positions of these CNV changes in DEGs are detailed in Fig 1E.

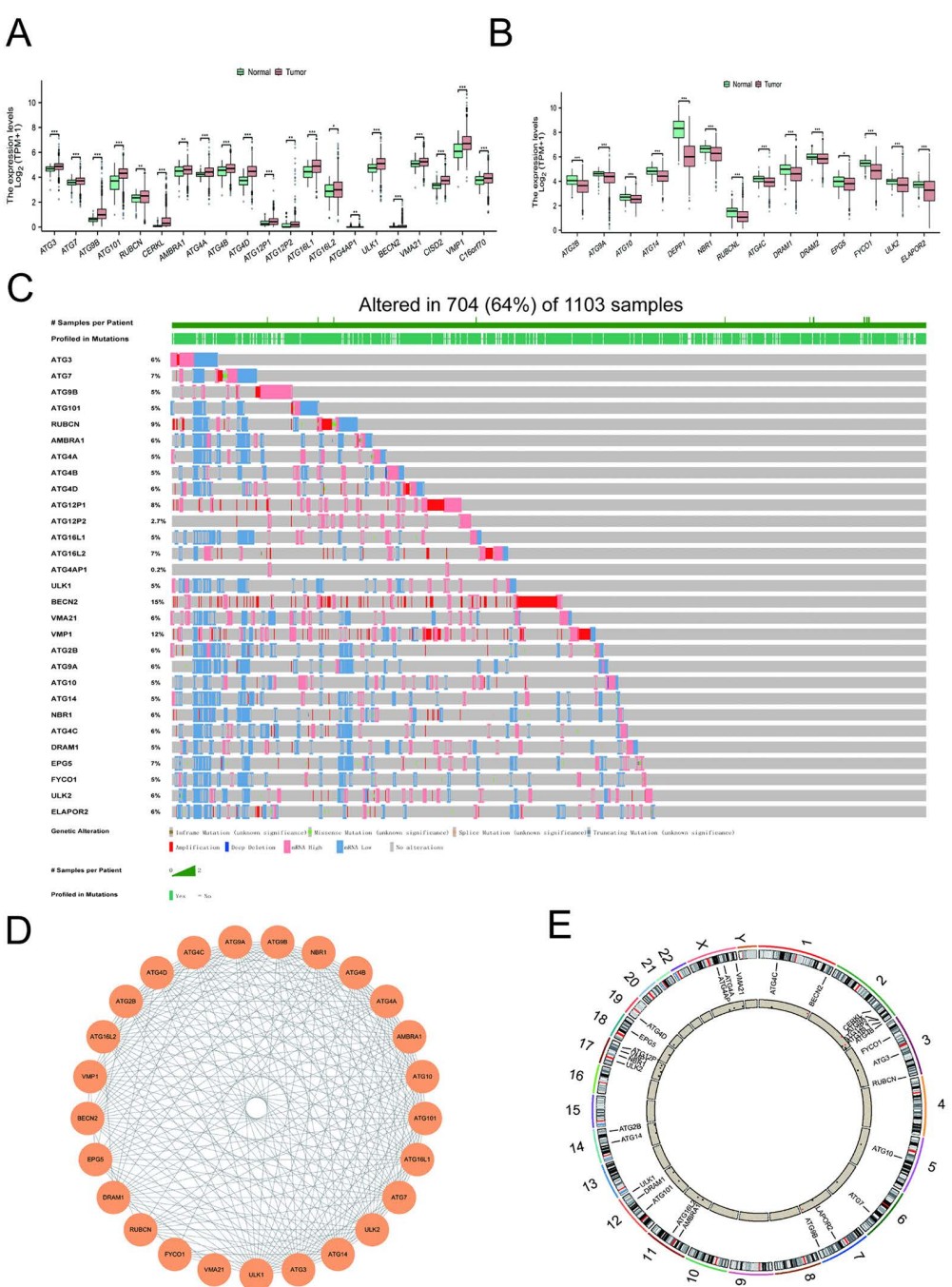

**Fig 1. Differential expression and genetic variation of DEGs in BRCA. (A)** The expression of 21 autophagy-related genes was upregulated in the breast cancer cohort; (B) 14 autophagy-related genes associated with downregulated expression in the breast cancer cohort; **(C)** The mutational water-fall plot of DEGs from TCGA-BRCA showed that BECN2, VMP1, and RUBCN had the highest mutation frequencies; **(D)** PPI network of DEGs generated using the STRING database; **(E)** Chromosomal locations of CNV alterations for 29 DEGs on 23 chromosomes. Note: DEGs: Cancer-related differentially expressed genes; BRCA: Breast cancer; CNV: Copy number variation. *$P < 0.05$; **$P < 0.01$; ***$P < 0.001$.

## Development and validation of an autophagy-related gene prognostic risk model in BRCA

Analysis of all autophagy-related genes was performed with an integrated machine learning pipeline. Following comprehensive evaluation based on mean AUC values at 1-, 3-, and 5-year intervals, the Elastic_net_0.1 model was identified as the optimal algorithm (Fig 2A). The Elastic_net_0.1 model demonstrated consistently high mean AUC values across all three time points (1-, 3-, and 5-year). A heatmap effectively visualized the regression coefficients of input genes in diverse prognostic models. Meta-analysis further validated the model's efficacy by comparing survival outcomes across multiple datasets. The Elastic_net_0.1 model was validated as a significant prognostic factor for diverse survival outcomes across multiple independent cohorts (Fig 2B). Fig 2C illustrates the Kaplan-Meier (KM) survival analysis of the Elastic_net_0.1 regression model across validation cohorts, demonstrating significantly poorer prognosis in high-risk score patients compared to low-risk counterparts ($P < 0.05$).

## RUBCN serves as a novel risk predictor in BRCA

In all models, RUBCN demonstrated significant positive coefficient consistency, indicating that higher expression levels contributed to increased risk scores, which suggests that RUBCN is a risk gene for BRCA (Fig 3A). Meta-analysis results revealed a pooled risk ratio of 1.41 (95% confidence interval: 1.23–1.61) for RUBCN, further confirming its role as a risk gene for BRCA (Fig 3B). To evaluate RUBCN expression at the protein level, we performed immunohistochemical (IHC) analysis on five paired samples of primary breast cancer and adjacent normal tissues. Quantitative IHC scoring revealed that RUBCN protein levels were significantly higher in tumor tissues compared with their paired normal counterparts (Fig 3C).

## Association of RUBCN with BRCA clinicopathological features

The prognostic nomogram incorporating RUBCN expression and clinicopathological variables, including age, pathological T stage, N stage, M stage, ER status, PR status, and HER2 status, demonstrated significant predictive accuracy for 1-, 3-, and 5-year survival rates in BRCA patients (Fig 4A). The calibration curves demonstrated strong agreement between the nomogram-predicted and observed survival rates at 1, 3, and 5 years (Fig 4B). RUBCN expression levels were significantly elevated in breast cancer tissues compared to normal breast tissues across all age groups, molecular subtypes, and TNM stages, though no statistically significant differences were observed among the TNM substages ($P > 0.05$; Fig 4C). Survival analysis revealed that elevated RUBCN expression was significantly associated with poorer overall survival (OS) in BRCA-mutated patients (Fig 4D). These findings underscore the importance of RUBCN as a prognostic biomarker in BRCA-associated cancers.

## Functional and pathway enrichment analysis

As shown in Fig 5, GO enrichment analysis revealed significant enrichment in Biological Process (BP), including negative regulation of endopeptidase activity, negative regulation of proteolysis, and hormone metabolic process (Fig 5A); Enrichment analysis for Cellular Component (CC) revealed predominant localization to secretory granule lumen, dense core granule, and collagen-containing extracellular matrix (Fig 5B); As for Molecular Function (MF), significant enrichment was observed in signaling receptor activator activity, G protein-coupled receptor binding, and chemokine receptor binding (Fig 5C). KEGG pathway analysis revealed significant enrichment in the neuroactive ligand-receptor interaction pathway, IL-17 signaling pathway, and metabolic regulation pathways (Fig 5D). GSEA revealed significant positive correlations between elevated RUBCN expression and pathways associated with cell cycle checkpoint, homologous recombination, DNA damage response, sister chromatid cohesion establishment, DNA repair, telomere maintenance, mitotic spindle assembly, and mismatch repair (Fig 5E). Pearson correlation analysis revealed significant positive correlations between RUBCN expression levels and gene set scores associated with BRCA, particularly in pathways implicated in DNA repair, DNA damage response, and tumor invasion (Fig 5F).

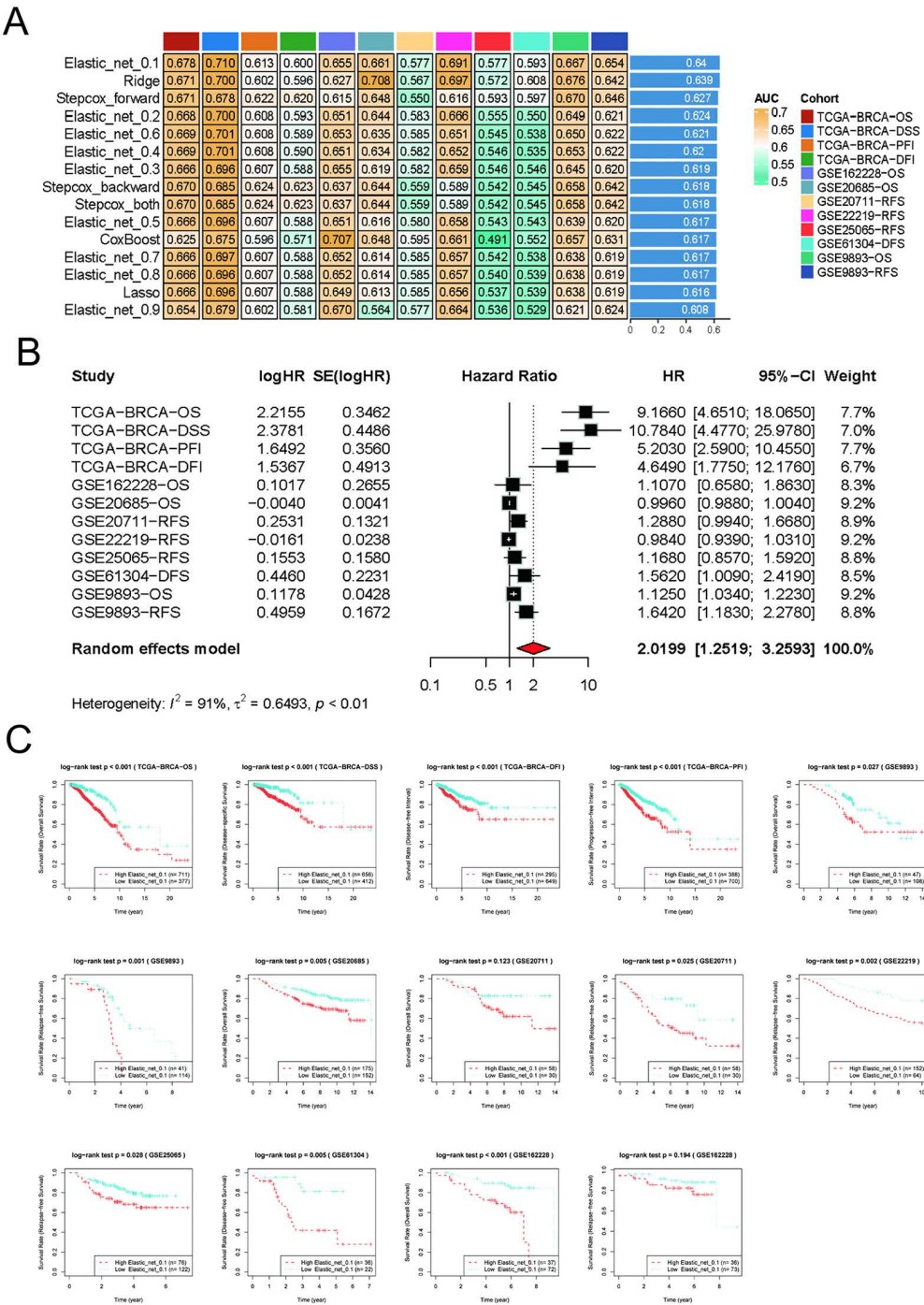

**Fig 2. Identification of key autophagy genes and development of a prognostic prediction model. (A)** Comparison of average AUC values at 1-, 3-, and 5-year intervals for prediction models developed using different algorithms. This heatmap compares the average AUC (Area Under the Curve) values across prediction models constructed with different algorithms at three time points: 1, 3, and 5 years; **(B)** Meta-analysis of hazard ratios in the prognostic model; **(C)** Kaplan-Meier survival analysis of the prognostic Elastic Net (α = 0.1) regression model.

## A

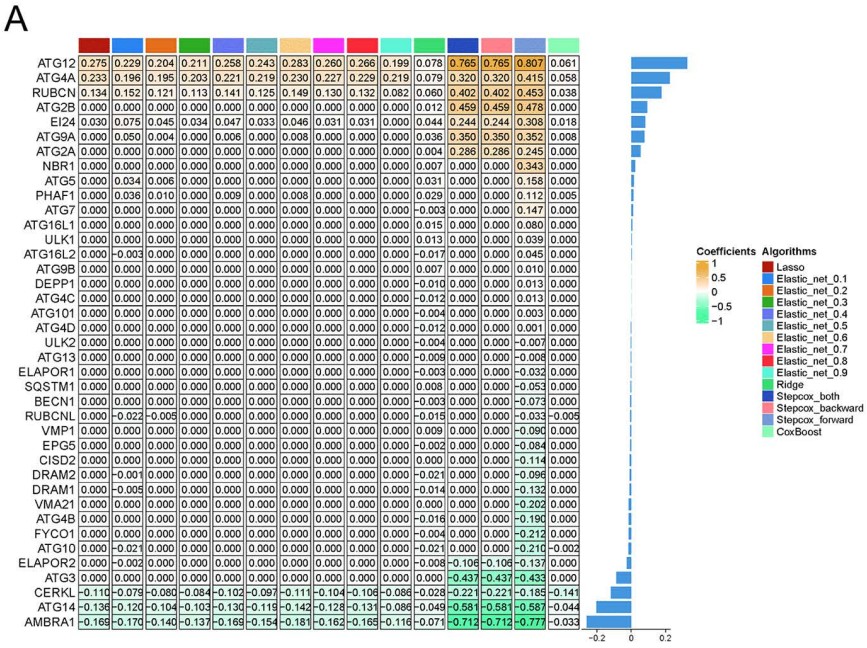

## B

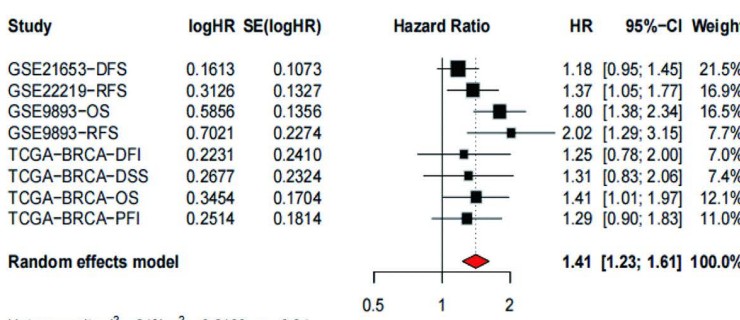

| Study | logHR | SE(logHR) | Hazard Ratio | HR | 95%-CI | Weight |
|---|---|---|---|---|---|---|
| GSE21653–DFS | 0.1613 | 0.1073 | | 1.18 | [0.95; 1.45] | 21.5% |
| GSE22219–RFS | 0.3126 | 0.1327 | | 1.37 | [1.05; 1.77] | 16.9% |
| GSE9893–OS | 0.5856 | 0.1356 | | 1.80 | [1.38; 2.34] | 16.5% |
| GSE9893–RFS | 0.7021 | 0.2274 | | 2.02 | [1.29; 3.15] | 7.7% |
| TCGA–BRCA–DFI | 0.2231 | 0.2410 | | 1.25 | [0.78; 2.00] | 7.0% |
| TCGA–BRCA–DSS | 0.2677 | 0.2324 | | 1.31 | [0.83; 2.06] | 7.4% |
| TCGA–BRCA–OS | 0.3454 | 0.1704 | | 1.41 | [1.01; 1.97] | 12.1% |
| TCGA–BRCA–PFI | 0.2514 | 0.1814 | | 1.29 | [0.90; 1.83] | 11.0% |
| **Random effects model** | | | | **1.41** | **[1.23; 1.61]** | **100.0%** |

Heterogeneity: $I^2 = 24\%$, $\tau^2 = 0.0108$, $p = 0.24$

## C

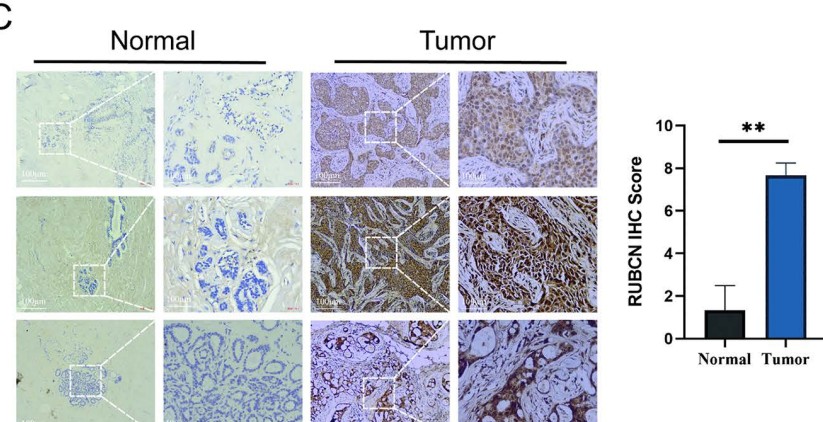

**Fig 3. Identifying and validating RUBCN as a key BRCA risk gene.** (A) shows the coefficients of each gene across multiple algorithms, where a coefficient of 0 means the gene wasn't used in the model; **(B)** Meta-analysis shows the pooled log-HR (log hazard ratio) value from univariable Cox survival analysis for RUBCN, indicating it's a BRCA risk gene; **(C)** Immunohistochemical staining of RUBCN in carcinoma tissues and adjacent non-tumor tissues (magnification: 100× and 400×; scale bar, 100 μm); The score (intensity + extent) is shown on the right. *$P < 0.05$; **$P < 0.01$; ***$P < 0.001$.

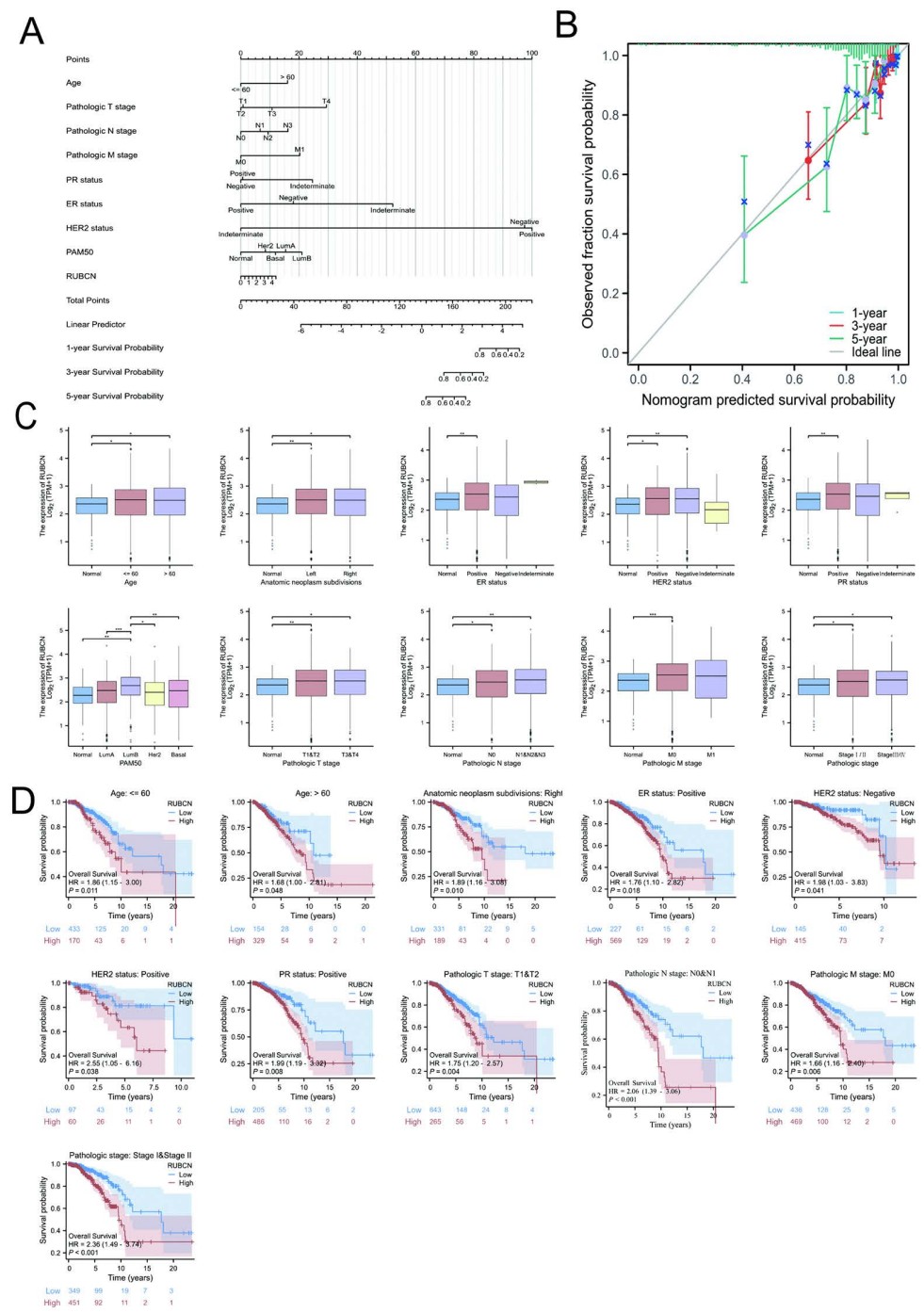

**Fig 4. Association of RUBCN expression with prognosis in breast cancer subtypes. (A)** Nomogram for predicting overall survival using clinical variables and RUBCN expression; **(B)** Calibration curves assessing the nomogram's accuracy in predicting 1-year, 3-year, and 5-year survival rates for breast cancer (BRCA) patients; **(C)** Visual depiction of correlations between RUBCN mRNA expression and key clinical factors in BRCA patients, including age, tumor location, ER status, PR status, HER2 status, Tumor stage (T), Metastasis stage (M), Node stage (N), and histological grade; **(D)** Kaplan-Meier curves comparing RUBCN expression levels and overall survival (OS) by clinical variables.

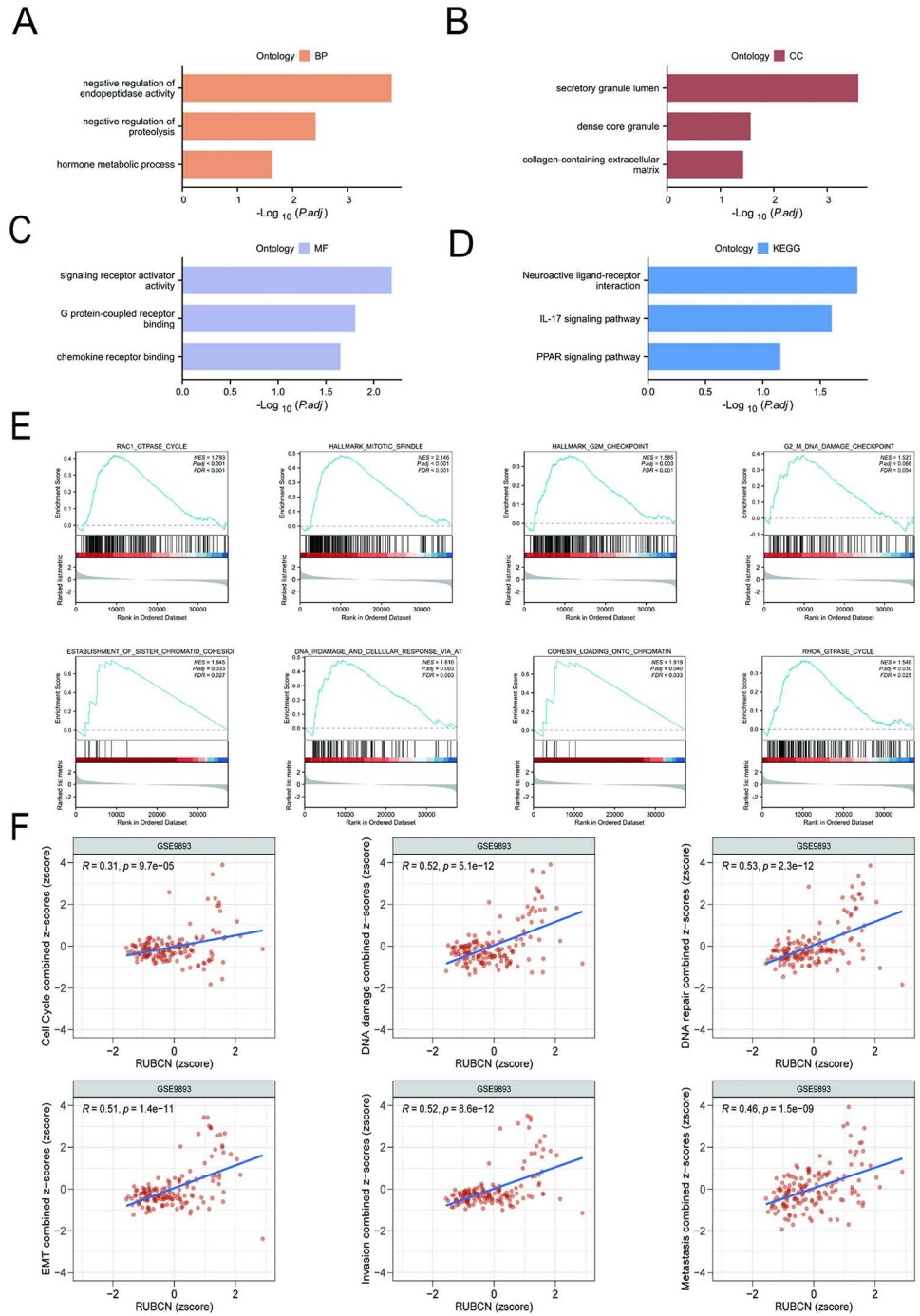

**Fig 5. Analysis of RUBCN function and pathway enrichment. (A)** The top 3 biological process terms from GO enrichment analysis of 450 DEGs; **(B)** The top 3 cellular component terms from GO enrichment analysis of 450 DEGs; **(C)** The top 3 molecular function terms from GO enrichment analysis of 450 DEGs; **(D)** KEGG pathway enrichment analysis indicated potential pathways; **(E)** Gene set enrichment analysis was used to evaluate and compare the enrichment scores of the indicated gene set in the high- and low-RUBCN¬expression groups in TCGA-BRCA. NES, normalized enrichment score; FDR, false discovery rate; **(F)** Analysis using the CancerSEA database revealed that RUBCN is critical for mediating multiple malignant phenotypes in BRCA, including cell cycle processes, DNA damage, DNA repair, invasion, and epithelial-to-mesenchymal transition.

## Spatially resolved expression of RUBCN predominates in malignant tumor niches

RUBCN expression was predominantly localized to malignant compartments. Spatial transcriptomics profiling of breast carcinoma sections revealed a significant positive correlation between RUBCN expression levels and the proportion of malignant cells within micro-regions, while showing an inverse association with stromal components, particularly immune cell infiltration. Furthermore, RUBCN expression gradually decreased from the tumor core to adjacent non-tumor tissues (Fig 6A–6H). These consistent spatial patterns indicate that dysregulated RUBCN expression and its biological impact within the tumor microenvironment are primarily attributable to malignant cell populations.

## Immune infiltration landscape associated with RUBCN in BRCA

Patients were stratified into high and low RUBCN expression cohorts, and the proportional distribution of 22 immune cell subsets was analyzed between these groups (Fig 7A). Comparative analysis of 22 immune cell subtypes between RUBCN high- and low-expression cohorts revealed significant reductions in CD8+T cells, natural killer (NK) cells, and follicular helper T (Tfh) cells within the high-expression group, whereas M2 macrophage infiltration was markedly elevated ($P < 0.05$; Fig 7B). Single-sample gene set enrichment analysis (ssGSEA) of the breast cancer tumor microenvironment revealed that among 24 immune cell types, the infiltration levels of 12 immune cells exhibited significant negative correlations with RUBCN expression. Notably, RUBCN expression positively correlated with central memory T-cell infiltration ($r = 0.524$, $P < 0.001$) and T-helper cell abundance ($r = 0.340$, $P < 0.001$). Conversely, plasmacytoid dendritic cells (pDCs) and natural killer (NK) cells showed pronounced negative correlations with RUBCN expression (both $P < 0.001$; Fig 7C). Stratification of 1,077 patients from the TCGA database by molecular subtypes (C1-C6) revealed pronounced enrichment patterns of RUBCN expression cohorts. The low-expression cohort ($n = 539$) showed highest prevalence in the C1 subtype (37%), whereas the high-expression cohort ($n = 538$) was robustly enriched in the C2 subtype (44%) ($P < 0.001$; Fig 7D). These findings demonstrate significant subtype-specific disparities in RUBCN expression across breast cancer immune subtypes.

## The effect of RUBCN knockdown on the proliferation, migration and invasion of breast cancer MDA-MB-231 cells

To further elucidate the definitive independent role of RUBCN in breast cancer, MDA-MB-231 cells exhibiting the highest RUBCN expression were selected for in vitroexperiments (Fig 8A, 8B). The correlation between RUBCN expression levels and in vitroproliferative, migratory, and invasive capacities was subsequently validated. RUBCN-knockdown MDA-MB-231 cells were generated, with knockdown efficiency confirmed by quantitative RT-qPCR and Western blotting (Fig 8C, 8D). Cell proliferation assessed via plate colony formation, CCK-8, and EdU incorporation assays demonstrated that RUBCN knockdown significantly suppressed the proliferation of MDA-MB-231 cells (Fig 8E–8G). Wound healing and Transwell assays revealed that RUBCN knockdown markedly attenuated both the migratory and invasive capacities of MDA-MB-231 cells (Fig 9A, 9D). Western blot analysis of LC3-II and p62 protein levels was performed to further elucidate RUBCN's role in regulating autophagic flux. Fig 9E shows that although RUBCN knockout altered the steady-state level of p62 under basal conditions, the chloroquine assay demonstrated a significant increase in the accumulation of LC3-II in RUBCN–knockout cells, revealing an enhancement in autophagic flux.

## Discussion

Breast cancer currently ranks as the most prevalent malignancy among women and is one of the leading causes of cancer-related mortality. According to 2022 reports, approximately 2.3 million new cases and 670,000 deaths occurred globally, imposing a substantial burden on healthcare systems [1]. Therefore, elucidating the molecular mechanisms underlying breast carcinogenesis and identifying novel therapeutic targets are of paramount importance for improving patient prognosis.

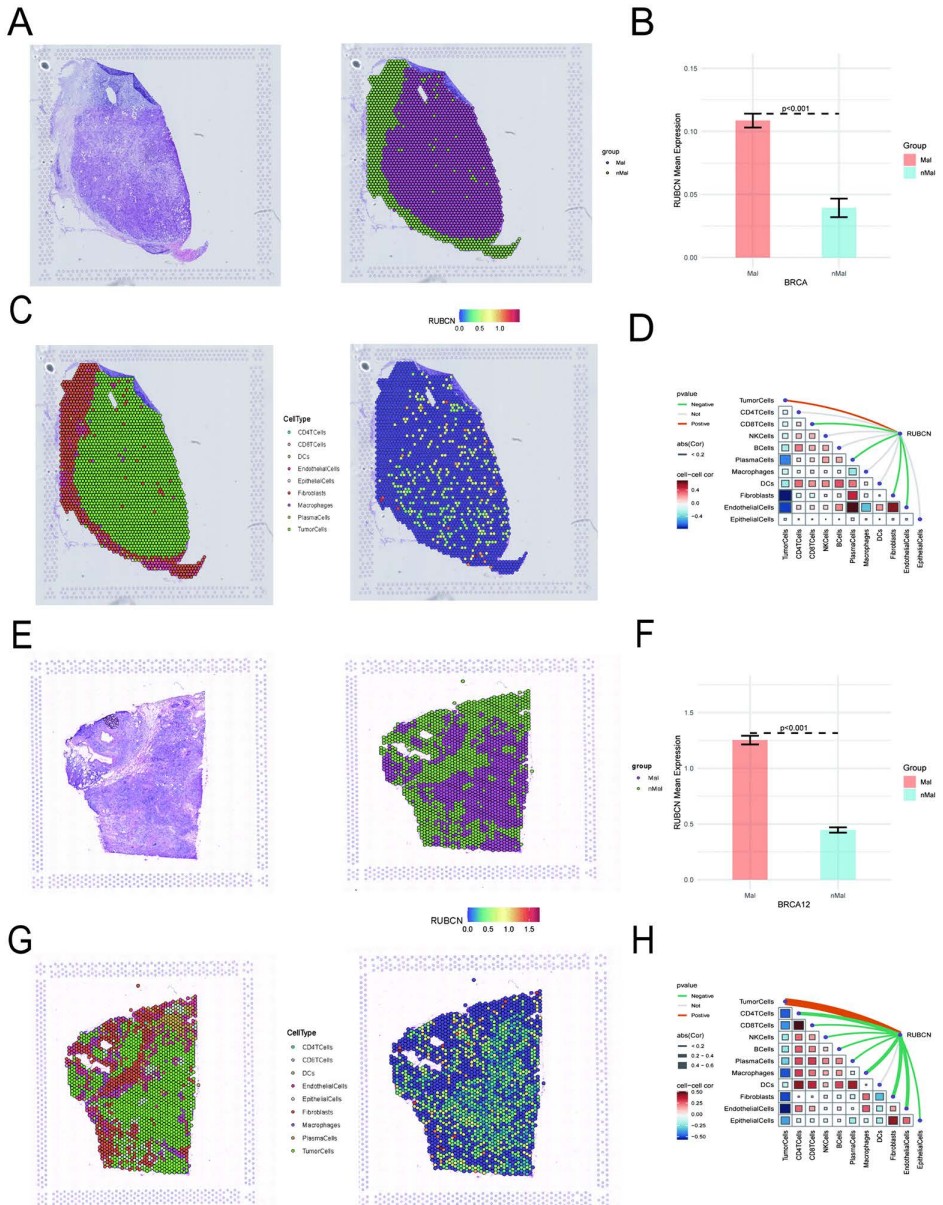

**Fig 6. Analysis of RUBCN expression and immune correlations in the tumor microenvironment using spatial transcriptomics. (A–H)** The images are derived from two cases of breast cancer, where Mal denotes malignant and nMal denotes non-malignant (tissue). **(A, E)**Tissue sections serving as blank controls, with tumor and normal tissue regions demarcated by distinct colors to indicate different microdomain types; **(B, F)** Microdomain differential analysis, where distinct groups are color-coded, and bar heights represent the mean expression levels per group; **(C, G)**Each scatter point represents a microdomain labeled by its most abundant cell type, with cell types color-coded.Individual points correspond to spatial transcriptomic sequencing spots, with deeper red indicating higher gene expression at that location; **(D, H)** Correlation analysis: red lines denote positive correlations, green lines negative correlations, and gray lines nonsignificant associations. Line thickness reflects the absolute value of the correlation coefficient. Triangular origins indicate correlation strength, represented by the intensity and size of colored squares (red: positive; blue: negative). Deeper hues and larger squares correspond to more significant p-values and higher absolute correlation coefficients, respectively.

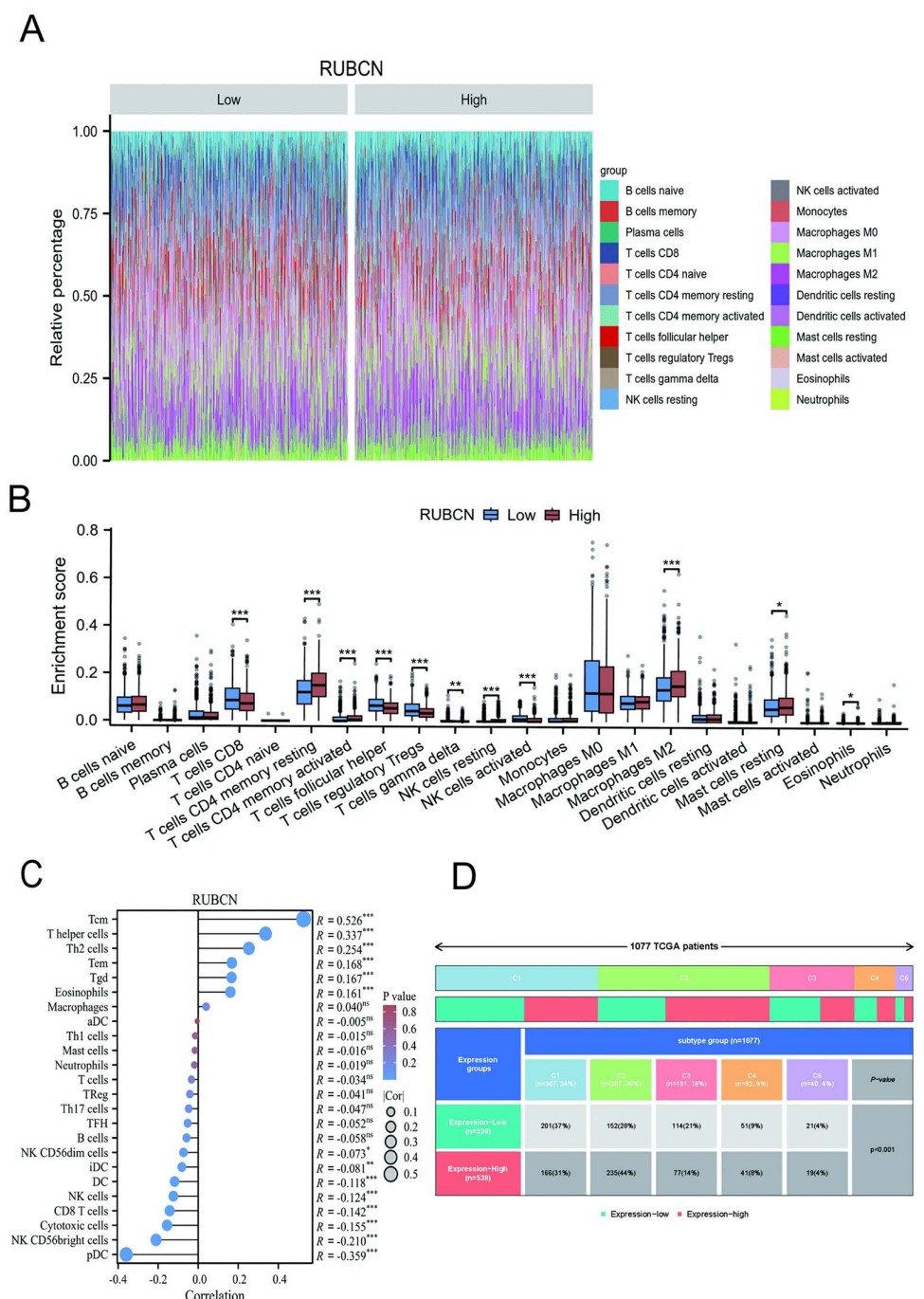

**Fig 7. RUBCN expression modulates immune microenvironment in breast cancer. (A)** Stacked bar chart comparing immune cell infiltration levels between high and low RUBCN expression; **(B)** RUBCN enrichment scores in different immune cells; **(C)** Lollipop plot showing the correlation between RUBCN expression and infiltration levels of 24 immune cell types; **(D)** Proportion of high/low RUBCN-expressing patients across distinct immune subtypes defined by cancer immune landscape classification, grouped by median RUBCN expression level, respectively. C1 (wound healing); C2 (IFN-gamma dominant); C3 (inflammatory); C4 (lymphocyte depleted); C6 (TGF-b dominant).

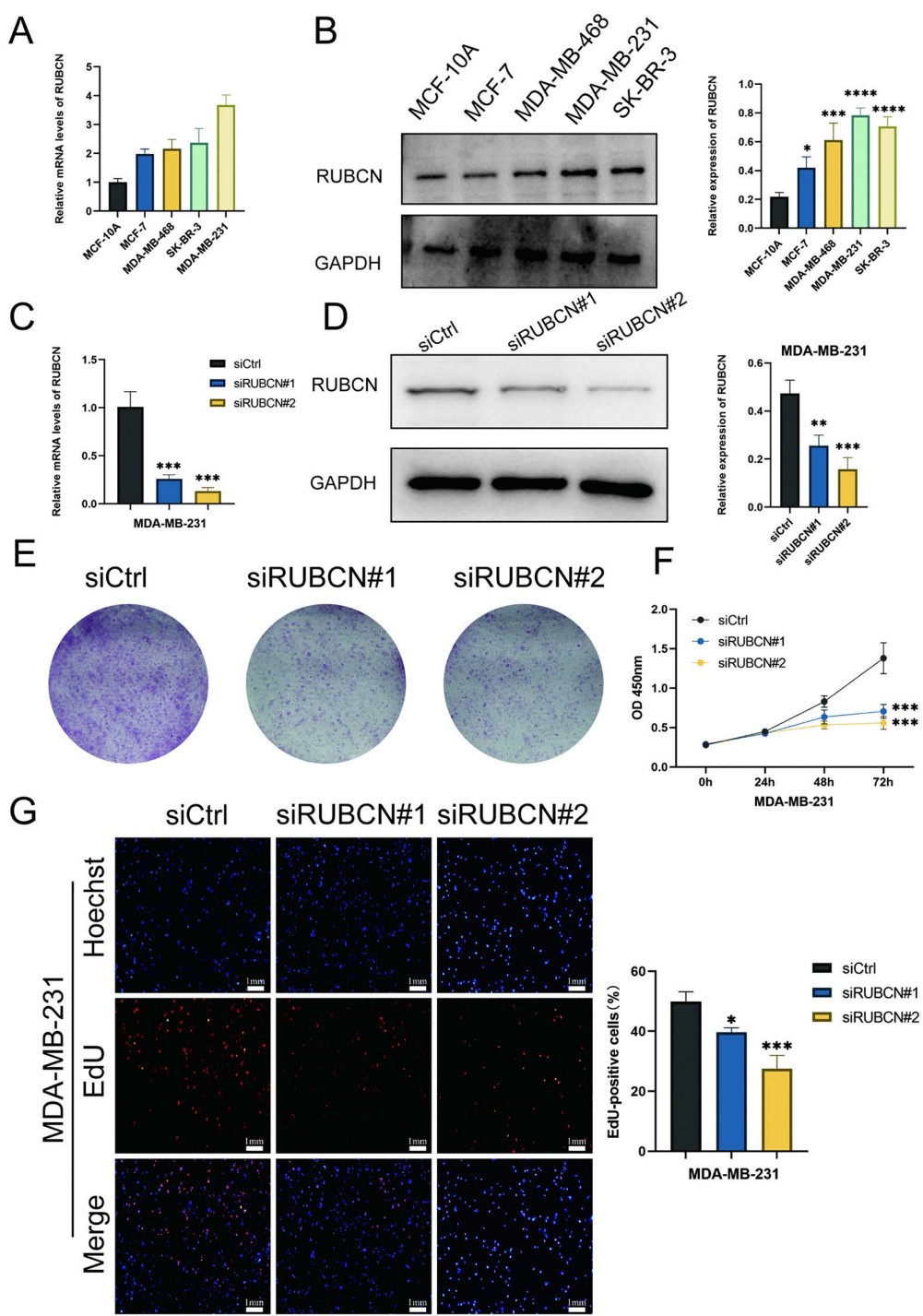

**Fig 8. Expression of RUBCN in breast cancer and its knockdown suppressing breast cancer cell proliferation. (A, B)** Expression of RUBCN in BRCA cell lines; **(C)** RUBCN expression detected by quantitative RT-PCR in control and RUBCN-knockdown cells; **(D)** Protein levels of RUBCN measured by Western blot in control and RUBCN-knockdown cells; **(E)** Colony formation assay of BRCA cells with RUBCN knockdown; **(F)** CCK-8 assay of BRCA cells with RUBCN knockdown; **(G)** Representative images and statistical analysis of EdU+ cells after RUBCN knockdown. *$P < 0.05$; **$P < 0.01$; ***$P < 0.001$. OD450, optical density at 450 nm.

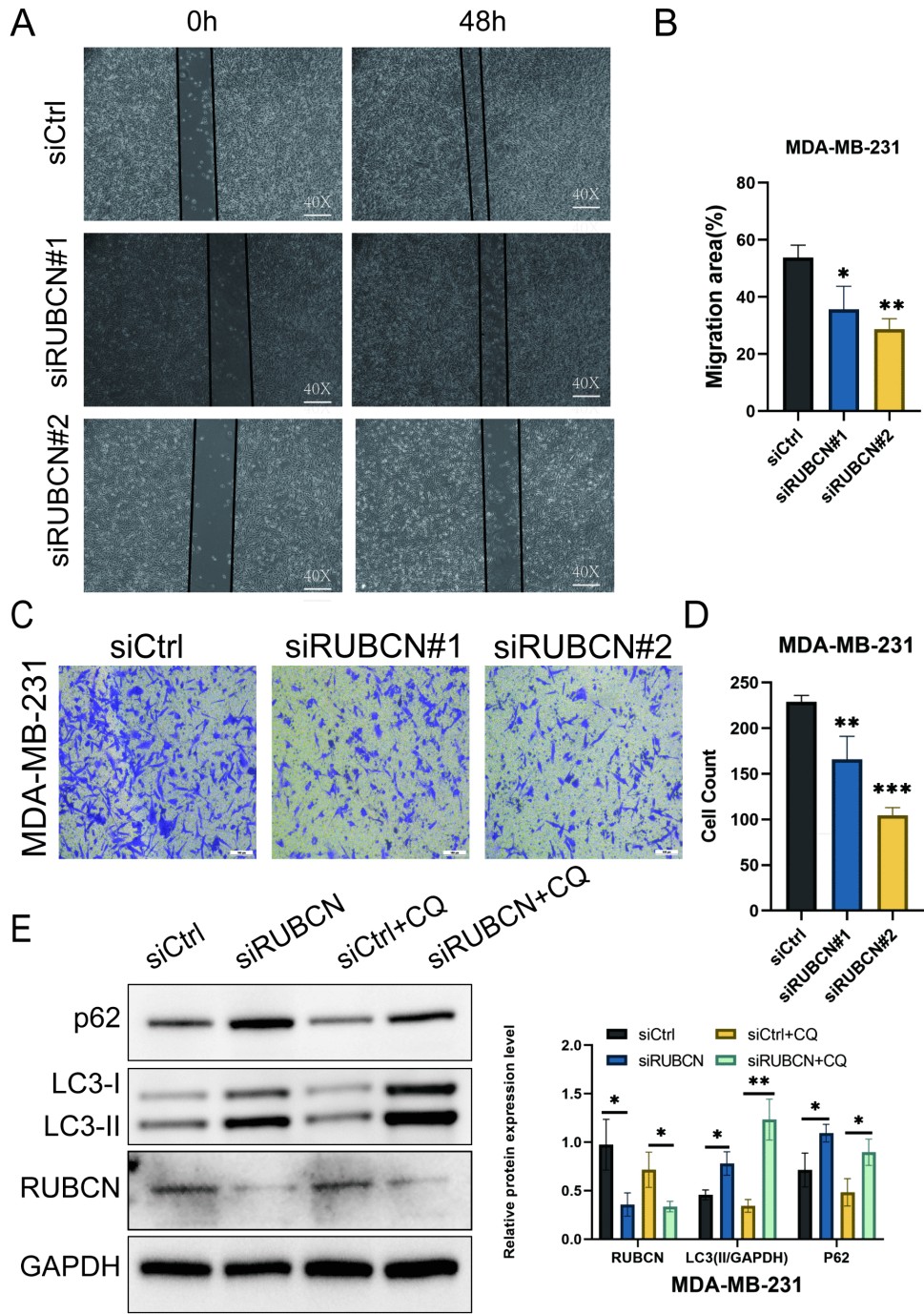

**Fig 9. RUBCN knockdown regulates migration, invasion, and autophagic flux in breast cancer cells. (A, B)** Knockdown of RUBCN affects the migration capability of BC cells. **(C, D)** Knockdown of RUBCN affects the invasion capability of BC cells. **(E)** Knockdown of RUBCN enhanced autophagic flux in cells. *$P < 0.05$, **$P < 0.01$, ***$P < 0.001$.

Recent studies have established that autophagy is essential for maintaining cellular homeostasis. Dysregulation of this process, however, exerts multifaceted roles in tumor initiation, progression, therapeutic response, and drug resistance [9]. In our study, multi-omics analysis revealed the critical role of ARGs in breast cancer. The Elastic_net_0.1 model identified RUBCN as a risk gene for breast cancer. Existing studies have demonstrated that inhibiting RUBCN expression arrests the cell cycle and suppresses tumor cell proliferation; however, its specific mechanism in breast cancer remains unclear [27], We demonstrated that RUBCN is significantly overexpressed in both cell lines and breast cancer tissues, with its elevated expression being strongly associated with poor prognosis as evidenced by TCGA and GEO datasets. Functional analyses revealed that knocking down RUBCN expression markedly reduced tumor cell proliferation, migration, and invasion capacities. Furthermore, our bioinformatic analysis revealed a significant negative correlation between high RUBCN expression and levels of immune cell infiltration (e.g., CD8+T cells and NK cells), alongside a positive correlation with M2 macrophage polarization. These intriguing associations suggest a potential link between RUBCN and an immunosuppressive tumor microenvironment; however, it is crucial to emphasize that these data are correlative and do not establish causality. The limited sample size, particularly within the TNBC subset, precluded a robust validation of this immune-related hypothesis. Therefore, the proposed mechanism by which RUBCN might modulate the immune landscape remains speculative and represents a key limitation of our study, requiring direct experimental validation in future research. Additionally, the differential expression of RUBCN across molecular subtypes (e.g., C1/C2) implies its utility as a subtype-specific biomarker, offering a potential therapeutic target for personalized treatment strategies.

At the mechanistic level, silencing of RUBCN significantly alters the autophagic status of cells, and its loss leads to a global acceleration of the autophagic process. Moreover, therapeutic resistance to PI3K inhibitors in breast cancer is commonly associated with mTORC1-dependent autophagic defects [28]. RUBCN likely potentiates metabolic vulnerabilities through similar mechanisms, thereby rationalizing combinatorial targeting strategies (e.g., metabolic modulators+immune checkpoint blockade) as actionable therapeutic opportunities [29].

These findings collectively establish RUBCN as a viable therapeutic target for breast cancer. Its demonstrated immunomodulatory function in tumor immune infiltration suggests potential applications in immunotherapy development or combination treatment approaches. Further research should investigate the molecular mechanisms through which RUBCN recruits immune cells and assess their clinical implications for patient survival and treatment response. This study enhances our comprehension of tumor microenvironment regulation while providing a framework for precision therapy development in breast cancer. These insights may ultimately inform improved clinical management strategies for breast cancer patients.

## Conclusion

This study discovered that autophagy-related genes exhibit elevated expression in BRCA. Through Elastic Net regression, a consensus prognostic model was constructed, identifying RUBCN as a core prognostic gene. Validation experiments confirmed significantly elevated RUBCN expression in BRCA tissues and cell lines. High RUBCN expression correlated significantly with poor patient prognosis. Furthermore, RUBCN expression showed a negative correlation with immune cell infiltration, suggesting its potential role in mediating tumor immune escape by suppressing immune infiltration, thereby promoting BRCA tumorigenesis. Functional assays demonstrated that RUBCN knockdown markedly suppressed proliferative, migratory, and invasive capacities of BRCA cells. Therefore, targeting RUBCN may represent a promising therapeutic strategy for BRCA.

## Supporting information

**S1 Fig. Dysregulation of core autophagy machinery in breast carcinogenesis (A-J) Systematic comparison of mRNA expression for 10 key autophagy regulators between normal breast tissue and invasive ductal carcinoma.** (PDF)

**S2 Fig. Differential mRNA expression of core autophagy-related genes in matched normal and tumor breast tissues.** (A-K) Paired comparative analysis reveals dysregulation of 11 key autophagy-related genes (ARGs) in breast cancer.
(PDF)

**S3 Fig. Differential mRNA expression of 18 core autophagy-related genes in paired normal and neoplastic breast tissues.** (A-Q) Paired box-and-whisker plots comparing transcript levels of 18 core autophagy regulators between histologically normal breast tissues and matched breast tumor specimens from the TCGA-BRCA cohort.
(PDF)

**S1 Table. Primers used in quantitative real-time polymerase chain reaction.**
(PDF)

**S2 Table. The primer sequences information of Rubcn siRNA.**
(PDF)

**S3 Table. Autophagy-related genes were curated from the HUGO Gene Nomenclature Committee (HGNC) database (https://www.genenames.org/).**
(PDF)

**S1 Data. Raw images.**
(PDF)

**S1 File. Original IHC data.**
(DOCX)

**S2 File. Original colony formation data.**
(DOCX)

**S3 File. Original CCK-8 data.**
(XLSX)

**S4 File. Original EdU data.**
(DOCX)

**S5 File. Original qRTPCR data.**
(XLSX)

**S6 File. Original qRTPCR data2.**
(XLSX)

**S7 File. Original transwell data.**
(DOCX)

**S8 File. Original cell migration data.**
(DOCX)

## Acknowledgments

The authors thank Zhiwei Jin for expert technical assistance in R programming.

## Author contributions

**Conceptualization:** Gui Lin Huang, Ji Xue Hou.

**Data curation:** Dong Dong Yang, Sheng Qiu Jia, Ming Ming Zhang.

**Formal analysis:** Dong Dong Yang, Cheng Hao Liu, Ze Kuan Xue, Ming Ming Zhang.

**Funding acquisition:** Ji Xue Hou.

**Project administration:** Gui Lin Huang, Ji Xue Hou.

**Resources:** Yong Zhou Huang, Xin Chun Zhao.

**Software:** Cheng Hao Liu.

**Supervision:** Gui Lin Huang, Ji Xue Hou.

**Validation:** Rui Yang.

**Visualization:** Rui Yang, Xin Chun Zhao.

**Writing – original draft:** Cheng Hao Liu, Ming Ming Zhang, Rui Yang, Yong Zhou Huang, Ji Xue Hou.

**Writing – review & editing:** Yong Zhou Huang, Xin Chun Zhao, Bao San Han, Sheng Dong Nie, Gui Lin Huang, Ji Xue Hou.

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
