## [Decision Letter · Decision Letter 0]

29 Oct 2025

Dear Dr. Huang,

We look forward to receiving your revised manuscript.

Kind regards,

Amy McCart Reed

Academic Editor

PLOS ONE

Journal Requirements:

“This work was supported by the National Natural Science Foundation of China (No.82260105), Project supported by the Science and Technology Program of Xinjiang Production & Construction Corps, China (No. 2024AB065) and Hospital-Level Research Project (No. BS202202).”

4. Regarding blot/gel data: PLOS ONE now requires that submissions reporting blots or gels include original, uncropped blot/gel image data as a supplement or in a public repository. This is in addition to complying with our image preparation guidelines described at https://journals.plos.org/plosone/s/figures#loc-blot-and-gel-reporting-requirements. These requirements apply both to the main figures and to cropped blot/gel images included in Supporting Information. If the manuscript is positively reviewed, we will ask the authors to provide any missing raw image data for blot/gel results when they submit their first revision. As part of your review, please ensure that figures reporting blot or gel images comply with the journal’s image preparation guidelines and that the original data are provided following the journal’s request.  If you have any questions or concerns about blot/gel figures or data for this submission, please email us at plosone@plos.org before issuing a decision letter.

Reviewers' comments:

Reviewer's Responses to Questions

**Comments to the Author**

1. Is the manuscript technically sound, and do the data support the conclusions?

Reviewer #1: Yes

Reviewer #2: Yes

2. Has the statistical analysis been performed appropriately and rigorously?

Reviewer #1: Yes

Reviewer #2: Yes

3. Have the authors made all data underlying the findings in their manuscript fully available?

Reviewer #1: Yes

Reviewer #2: Yes

4. Is the manuscript presented in an intelligible fashion and written in standard English?

Reviewer #1: Yes

Reviewer #2: Yes

Reviewer #1: Overall comment:

The authors have done a good job exploring RUBCN as a prognostic biomarker and potential therapeutic target in breast cancer. The results are clear and concise, and the discussion is well-justified. A few points that could be considered to further enhance the quality of the paper are listed below:

INTRODUCTION:

Flow of the narrative: The introduction could benefit from smoother transitions as the topic shifts. For example, lines 75–77 mention that regulated cell death (RCD) pathways have been characterized, but it is unclear in what context. Are these pathways characterized in cancers? If so, specifying the types of cancers would strengthen the introduction. It would also be useful to briefly provide your own inference rather than relying solely on the reference, especially as the paper focuses on an autophagy-related gene.

Autophagy in cancer: line 81, the discussion of autophagy in cancer could be more contextualized. The statement “Autophagy exerts tumor suppressive effects by clearing dysfunctional organelles to maintain genomic integrity, conversely facilitating tumor cell survival under nutrient deprivation” could be clarified. If the intention is to highlight the dual role of autophagy, it would be helpful to explicitly state this or rephrase the sentence for clarity.

Transition between ATG4A and RUBCN: The shift from discussing ATG4A to RUBCN feels abrupt and difficult to follow. Consider creating a smoother transition that clearly links the discussion of ATG4A to the relevance of RUBCN in the context of autophagy and cancer.

RESULTS:

Figure 1c: Highlighting the genes of interest (BECN2, VMP1, and RUBCN) would help the reader focus on the key points being discussed.

Figure 3c: When comparing expression between normal breast tissue and breast cancer, the representative images are predominantly stromal and lack clear breast structures such as ducts. Including images with more cellular regions could better demonstrate the cancer cell–specific expression of RUBCN.

Discussion:

The authors state: “Furthermore, RUBCN was found to potentially accelerate tumor progression by suppressing antitumor immunity (e.g., reducing CD8+ T cell and NK cell infiltration) while promoting an immunosuppressive microenvironment (e.g., increasing M2 macrophage polarization).”

It is not clear that a negative correlation alone can be interpreted as a functional effect. Based on the data presented, it may be premature to conclude that RUBCN accelerates tumor progression. Additionally, the specific breast cancer subtypes included in this analysis are not stated; if subtypes were included, this should be clearly indicated and contextualized, as the immune microenvironment can vary substantially between subtypes.

Reviewer #2: This is a well-structured and logical study following a standard and appropriate workflow for biomarker discovery and functional validation. The overall narrative is coherent, and the findings are of potential translational relevance. However, several revisions are recommended to improve clarity, completeness, and rigor before the manuscript can be considered for publication.

Introduction:

The second paragraph feels slightly displaced and interrupts the flow. I suggest streamlining this section to ensure the knowledge gap and rationale are clearly articulated. The purpose of the study should be stated more explicitly to avoid it being lost.

Line 100 – RUBCN how will it be used to diagnose? Consider removing and keeping for treatment.

Clinical and subtype relevance: While the functional assays were conducted in a TNBC cell line (Fig. 4), no TNBC-specific clinical data from TCGA are provided to support this model choice.

→ Please stratify TCGA analyses by molecular subtype, particularly TNBC, to verify that the observed associations are maintained in this relevant subgroup.

→ If sample size is limiting, this can be acknowledged in the Discussion.

Autophagy knockdown interpretation: The knockdown approach is appropriate and adds mechanistic value; however, the current assessment of autophagy remains limited. Although migration, invasion, and proliferation assays were performed, assessing autophagy could provide valuable mechanistic insight into whether these phenotypic changes are driven by altered autophagic flux and could also help link the findings to potential immune-related effects.

→ I strongly recommend including a protein-level validation for example Western blot analysis of LC3-I/II and p62/SQSTM1 to confirm autophagy modulation. Even without a full flux assay, this would provide critical mechanistic support. Alternatively, the addition of a basic immune-related readout (e.g. CD8A, CXCL9/10 expression or similar) could also help to reinforce the proposed immunological relevance (in the discussion) or a marker pulled down from fig 5?

Methods Clarifications:

• Line 210: Section title reads like a Results heading. Suggest renaming (e.g., “Spatial transcriptomic profiling of the breast cancer TME”) and moving to line 212.

• Please clarify how many spatial transcriptomic cases were analysed.

• Immunohistochemistry: specify formalin percentage, antigen retrieval temperature, and duration. Imaging specifications missing. Any quantification of IHC??

• Line 239–240: “The authors declare no conflicts of interest” is misplaced in the Methods section.

• Western blot: please state that experiments were performed in triplicate (visible in supplementary but not stated in text).

• How long were cells kept before quantifying EdU? Also please specify how EdU was imaged and quantified? Was it normalised by hoechst?

• How was fig 9D quantified, no units on x axis.

• Ensure catalogue numbers (cat#) are listed consistently throughout all methods.

Results & Figure-Specific Feedback:

Figure 3: A PCA plot may be more informative than a heatmap to illustrate sample separation.

→ Additionally, representative IHC images (Fig. 3c) appear predominantly stromal; include more epithelial rich regions with clearer ductal structures. Mention scale bar in figure legend (fig3). Tumour IHC representative images has high background, lots of stromal labelling. Hard to see any negative tissue.

Line 337: Please indicate the number of genes included in the machine learning integration, was this the 35 DEGs? Also clarify criteria for classification into high vs low elastic net groups.

Figure 6:

→ Ensure plots are consistent in size and formatting.

→ Clearly label cell types and include a key for all heatmaps.

→ Panel E appears visually incomplete, please review.

→ The rationale for each spatial case should be better contextualised in the text.

The purpose of this figure doesn’t match how it is written under the results, needs better integration.

Figure 8: Could be moved to Supplementary, as it primarily establishes model setup. Need scale bars on all images. 8E images look blurry. 8F has heading ‘Data 3,’ please remove.

Please make all figures, graphs and images the same size and aligned. Make sure to include units of measurement on all graphs, just double check each one.

Discussion:

The statement that “RUBCN accelerates tumor progression by suppressing antitumor immunity…” overstates what can be concluded from correlative data. A negative correlation alone does not establish causation especially since no functional validation of the immune landscape was performed.

Subtype-specific immune context should be reported if applicable, immune microenvironment varies profoundly between ER+, HER2+, and TNBC.

The Discussion is strong overall but would benefit from deeper exploration of clinical implications, e.g.:

→ potential role of RUBCN as a predictive biomarker or therapeutic target

→ future relevance for immunotherapy responsiveness (??) or patient stratification (??)

Limitations need to be discussed.

If possible, consider including (even briefly) any clinicopathological correlations from your IHC data (e.g., stage, grade, subtype). This is not essential, but it would add meaningful translational strength.

**Do you want your identity to be public for this peer review?** For information about this choice, including consent withdrawal, please see our Privacy Policy

Reviewer #1: No

Reviewer #2: No

---

## [Author Response · Author response to Decision Letter 1]

26 Nov 2025

Dear Dr. Amy McCart Reed and Reviewers,

We wish to express our sincere gratitude for the opportunity to revise our manuscript and for your consideration of it. We greatly appreciate the time and effort invested by the editor and the reviewers. Their insightful and constructive comments have been invaluable in helping us significantly improve the quality and rigor of our work.

We have carefully addressed all the points raised in the reviewers' comments. Our point-by-point responses are detailed below, where we have outlined all the modifications made. The corresponding changes have been incorporated into the revised manuscript and are highlighted using the "Track Changes" feature.

Part 1: Response to Journal Requirements

We have thoroughly reviewed and ensured that our manuscript adheres to all formatting guidelines specified by PLOS ONE. The following items have been addressed:

1. Format template: The manuscript has been revised in accordance with the PLOS ONE formatting template provided in the link.

2. Code sharing: The code used in this study has been uploaded to GitHub. The relevant information has been provided in the Data Availability Statement.

3. Funding Statement: Regarding the funder statement, we have updated the corresponding section in the Cover Letter as follows: "The funders had no role in the study design, data collection and analysis, decision to publish, or preparation of the manuscript."

4. Western Blot Data: We have prepared the full, uncropped original images for all Western blot experiments.

5. References: We have verified all references to ensure they are complete, accurate, and that no retracted papers have been cited.

Reviewer #1:

Comment 1: The narrative flow of the Introduction section could be enhanced. For instance, the description of the RCD pathway and the dual role of autophagy could be clearer, and the transition from ATG4A to RUBCN is somewhat abrupt.

Author's reply: We thank the reviewer for this valuable suggestion. Accordingly, we have rewritten the Introduction section to enhance its logical flow.

Location of the changes: The second paragraph of the Introduction section.

Comment 2: Key genes (BECN2, VMP1, RUBCN) in Figure 1c should be highlighted to help readers focus on the critical points under discussion.

Author's reply: We thank the reviewer for this excellent suggestion. In response, we have now explicitly labeled the three key genes (BECN2, VMP1, RUBCN) in the legend of Figure 1c to help readers more easily identify and focus on these critical elements.

Location of the changes: The legend for Figure 1c (Page 18, Lines 368-369).

Comment 3: The representative IHC images in Figure 3c are predominantly stromal and lack clear mammary structures, such as mammary ducts. They should include images of more cellular regions to better demonstrate the cancer cell-specific expression.

Author's reply: We have replaced the representative IHC images in Figure 3c. The new images display clearer mammary duct structures and cancer cell-enriched areas, thereby more effectively illustrating the differential expression of RUBCN between cancerous and normal tissues.

Location of the changes: Fig 3.

Comment 4: The conclusion that RUBCN accelerates tumor progression by suppressing anti-tumor immunity may be somewhat premature and needs to be stated more cautiously. Furthermore, the specific breast cancer subtypes included in the analysis should be clearly specified.

Author's reply: We thank the reviewer for raising this important point. We agree that correlation does not directly imply causation. Accordingly, we have modified the relevant statements in the Discussion section (Page 28, Lines 577-586), changing the wording from a causal claim to a description of the observed correlation. In addition, we have now noted as a limitation of our study that the findings warrant validation across various molecular subtypes in the future.

Location of the changes: In the Discussion section.

Reviewer #2:

Comment 1: The second paragraph of the Introduction could be more concise, and the research objectives should be stated more explicitly.

Author's reply: We have streamlined and rewritten the introduction by removing redundant information, thereby more clearly articulating the study's objectives and the knowledge gap it aims to address.

Location of the changes: In the Introduction section.

Comment 1: How could RUBCN be used in diagnosis? Consider removing this section and conserving the content for the discussion on therapeutic applications.

Author's reply: We sincerely thank the reviewer for this astute observation. We completely agree that the discussion regarding the diagnostic potential of RUBCN was premature in the absence of specific diagnostic data. Following the reviewer's suggestion, we have removed the entire section speculating on diagnostic applications from the manuscript. The discussion is now focused and strengthened, emphasizing the therapeutic implications of our findings. We appreciate the reviewer's guidance, which has helped us present a more precise and rigorous interpretation of our results.

Location of the changes: Page 6, Lines 103-106.

Comment 2: It is recommended to perform a subgroup analysis of the TCGA data focusing on TNBC, and to provide clinical data support for the functional experiments conducted in TNBC cell lines.

Author's reply: We thank the reviewer for this valuable suggestion. Due to the limited sample size of TNBC cases in the TCGA cohort, we have acknowledged this as a limitation of our study in the Discussion section (Page 28, Lines 582-586), emphasizing the need for future validation in larger TNBC cohorts. Furthermore, we have added a justification for selecting TNBC cell lines (specifically MDA-MB-231) for our functional experiments, noting their higher expression levels, more aggressive phenotype, and limited treatment options, which collectively underscore the clinical relevance of investigating novel targets in this subtype.

Location of the changes: In the Discussion section.

Comment 3: It is strongly recommended to include additional Western blot analyses (e.g., for LC3-II and p62) to validate the changes in autophagic flux, thereby strengthening the mechanistic interpretation.

Author's reply: We sincerely thank the reviewer for this critical suggestion. Following the recommendation, we have performed the additional experiments. Upon RUBCN knockdown, we assessed the protein levels of LC3-II and p62 by Western blotting. The results revealed an increase in both LC3-II and p62 levels, indicating enhanced autophagic flux upon RUBCN knockdown. These new data provide crucial mechanistic support for our phenotypic observations regarding proliferation, migration, and invasion. The corresponding results have been included in the revised manuscript as Figure 9E and its accompanying description.

Location of the changes: A new paragraph describing the results and Figure 9E has been added to the Results section, and the details of the Western blot experiment have been updated in the Methods section.

Comment 4: The Methods section requires further clarification regarding specific experimental details, such as the number of spatial transcriptomics samples, the procedures for IHC and EdU assays, and the methodology used for image quantification.

Author's reply: We have thoroughly revised the Methods section to include all missing experimental details.

1.Specifically, the number of cases included in the spatial transcriptomics analysis has been clearly stated.

2.Detailed IHC experimental conditions as well as image acquisition and quantification methods have been added.

3.Detailed specifications for the EdU assay have been provided, including the incubation time, imaging methodology, and quantification process.

4.We agree with the reviewer and have corrected this error. The conflict of interest statement has been removed from the Methods section and is now appropriately placed as required by the journal's guidelines. Thank you for highlighting this.

5.We thank the reviewer for highlighting the need for this important methodological detail. As suggested, we have now explicitly stated in the main text of the Methods section that all Western blot experiments were performed in triplicate. The corresponding data from these independent replicates are included in the supplementary materials.

6.We thank the reviewer for the comment. Figure 9D was quantified by counting the number of invaded cells. The x-axis represents the experimental groups (siCtrl vs. siRUBCN) and therefore has no unit, which we have now clarified in the figure legend. The corresponding quantification method has been added to the Methods section.

7.We confirm that all catalog numbers have now been consistently listed in the Methods section.

Location of the changes: The relevant subsections within the Materials and Methods section.

Comment 5: The reviewer has provided specific suggestions for improving the presentation and labeling of several figures.

Author's reply: We fully appreciate the reviewer's intention to further demonstrate inter-sample heterogeneity through PCA. However, the objective of this specific part of our study was primarily to illustrate the feature importance (i.e., model coefficients) of candidate autophagy-related genes, as determined by multiple machine learning algorithms. Figure 3A, as a heatmap, is specifically designed to visualize the distribution and consistency of these coefficients across different algorithms during the feature selection process, rather than for sample clustering.

Therefore, we have retained the heatmap as the optimal format for presenting feature importance. Regarding the number of genes, 7 out of the initial 46 genes were automatically excluded by the algorithmic filtering process for the following reasons:

1. Their expression levels were extremely low (close to zero or constant) in the public training dataset;

2. They contained missing values in a subset of samples, making them unsuitable for feature calculation;

3. They were zeroed-out by the model's regularization process.

Consequently, the final model incorporated 39 genes, whose coefficients are visualized. This filtering is an inherent part of robust feature selection and does not compromise the overall stability or predictive performance of the model.

The IHC images in Figure 3c have been updated to include clearer epithelial regions and ductal structures. A scale bar has been added to the figure legend, and the image contrast has been optimized to better distinguish specific staining from background. These changes have been implemented in the revised manuscript.

We have clarified the criteria for classifying the high- and low-elasticity network groups in the Materials and Methods section (Page 9, Lines 163-165).

Comment 6: Specific suggestions were provided to improve the presentation and labeling of multiple figures (e.g., Figure 6), including ensuring formatting consistency, adding clear labels, refining panel 6E, and strengthening the logical integration between the figures and the text.

Author's reply: We sincerely thank the reviewer for these meticulous and valuable suggestions. We have thoroughly revised Figure 6 according to the recommendations provided.

We thank the reviewer for these important technical comments. We have now added scale bars to all relevant images. The specific image in panel 8E has been re-acquired to ensure clarity, and the label "Data 3" in panel 8F has been removed as suggested. These corrections have been implemented in the revised figure.

We sincerely thank the reviewer for their insightful and constructive comments, which have significantly strengthened the discussion of our findings. We have carefully addressed each point as follows:Regarding the causal interpretation of the RUBCN-immune interface, we have revised the relevant statements in the Discussion section (Page 28, Lines 577-586). The wording has been changed from a causal assertion to a description of the observed correlations, replacing the claim that "RUBCN accelerates tumor progression by suppressing antitumor immunity" with a more precise and nuanced description of the observed negative correlations. We have explicitly stated that the exact mechanistic relationship requires future functional validation of the immune landscape.Concerning clinical implications and limitations, we have expanded the discussion to include the study's limitations and to explore more deeply the potential clinical significance of RUBCN, including its possible role as a predictive biomarker or therapeutic target.We believe these revisions substantially enhance the balance, clarity, and translational relevance of the Discussion section.

---

## [Decision Letter · Decision Letter 1]

19 Dec 2025

Dear Dr. Huang,

Thank you for submitting your manuscript to PLOS ONE. After careful consideration, we feel that it has merit but does not fully meet PLOS ONE’s publication criteria as it currently stands. Therefore, we invite you to submit a revised version of the manuscript that addresses the points raised during the review process.

We look forward to receiving your revised manuscript.

Kind regards,

Amy McCart Reed

Academic Editor

PLOS One

Journal Requirements:

Reviewers' comments:

Reviewer's Responses to Questions

**Comments to the Author**

Reviewer #1: All comments have been addressed

Reviewer #2: All comments have been addressed

2. Is the manuscript technically sound, and do the data support the conclusions?

Reviewer #1: Yes

Reviewer #2: Yes

3. Has the statistical analysis been performed appropriately and rigorously?

Reviewer #1: Yes

Reviewer #2: Yes

4. Have the authors made all data underlying the findings in their manuscript fully available?

Reviewer #1: Yes

Reviewer #2: Yes

5. Is the manuscript presented in an intelligible fashion and written in standard English?

Reviewer #1: Yes

Reviewer #2: Yes

Reviewer #1: The authors have done a great job in addressing all the comments and updating the manuscript accordingly. One last thing that needs to be added is in figure 9E, where the western blot is done on protein extracted from RUBCN KD MDAMB231s but I cannot see RUBCN expression as part of that blot- I think it's important to have an additional blot from this experiment for RUBCN expression for a complete story.

Reviewer #2: Thank you for carefully addressing my previous comments. I have a few remaining minor points and clarifications that would further strengthen the manuscript.

At line 68, please add “basal-like” after TNBC for clarity. Throughout the manuscript, please also remove all “—” symbols, as these appear to be auto-generated characters rather than standard punctuation.

At line 244, the Methods state that non-specific binding was blocked with 5% bovine serum albumin for 30 minutes; please clarify what the 5% BSA was diluted in. At line 301, please specify the manufacturer of Abmart.

For Figure 3C, it would be helpful to include representative images demonstrating staining intensity scores (0–3+). Please also clarify whether the graph represents a combined score and, if possible, provide a breakdown of staining intensity across normal and tumour tissues. In addition, please confirm that all images are at 100× magnification, as there appear to be differences in cell size across panels.

In the Results section (lines 398–400), when presenting immunohistochemistry data, please use caution when describing “upregulated expression” and clearly explain how the quantitative IHC results were derived.

Finally, please include the RUBCN antibody validation using knockout controls in Figure 9E, alongside the western blot analysis of LC3-II and p62, to further support the specificity and robustness of these findings.

**Do you want your identity to be public for this peer review?** For information about this choice, including consent withdrawal, please see our Privacy Policy

Reviewer #1: No

Reviewer #2: No

---

## [Author Response · Author response to Decision Letter 2]

24 Dec 2025

Title of the manuscript: RUBCN as a Novel Prognostic Biomarker and Therapeutic Target in Breast Cancer

Manuscript Number: PONE-D-25-46002

Dear Dr. Amy McCart Reed and Reviewers,

We are pleased to submit the revised version of our manuscript titled “RUBCN as a Novel Prognostic Biomarker and Therapeutic Target in Breast Cancer”. We sincerely thank you for handling our manuscript and for the constructive comments provided by the reviewers. We have carefully addressed all points raised in the previous round of review. The revisions have been made accordingly in the manuscript, and our point-by-point responses are detailed below. We believe the manuscript has been significantly improved and hope it now meets the journal’s standards for publication. Thank you for your time and consideration.

Reviewer #1:

We thank the reviewer for the valuable feedback. All points raised have been addressed below, and we believe these changes have strengthened the manuscript.

Comment #1: One last thing that needs to be added is in figure 9E, where the western blot is done on protein extracted from RUBCN KD MDAMB231s but I cannot see RUBCN expression as part of that blot- I think it’s important to have an additional blot from this experiment for RUBCN expression for a complete story.

Author's reply: We sincerely thank the reviewer for the positive feedback and for raising this important point. To provide a complete and rigorous validation of the knockdown model in this experiment, we have now performed the additional Western blot to assess RUBCN expression using the same protein samples from RUBCN-knockdown MDA-MB-231 cells. As suggested, the new RUBCN blot has been added to Figure 9E. This addition strengthens the figure by directly confirming the knockdown efficiency in the precise samples used for assessing autophagic flux, thereby providing a more complete story as suggested.

We are grateful for this insightful suggestion, which has undoubtedly improved the clarity and validity of our data.

Reviewer #2:

We thank the reviewer for the positive assessment of our revisions and for these additional constructive suggestions, which have helped us further improve the clarity and robustness of the manuscript. We have addressed all points as detailed below.

Comment #1: At line 68, please add “basal-like” after TNBC for clarity. Throughout the manuscript, please also remove all “—” symbols, as these appear to be auto-generated characters rather than standard punctuation.

Author's reply: We thank the reviewer for the suggestion. We have added "basal-like" after "TNBC" on line 68 for better clarity. Additionally, as recommended, we have removed all instances of the long dash ("—") throughout the manuscript and replaced them with standard punctuation.

Key locations of these changes include: Page 3, Lines 36-38; Page 5, Lines 70-73; Page 20, Lines 417-418.

Comment #2: At line 244, the Methods state that non-specific binding was blocked with 5% bovine serum albumin for 30 minutes; please clarify what the 5% BSA was diluted in. At line 301, please specify the manufacturer of Abmart.

Author's reply: We apologize for the omission. The 5% BSA was diluted in phosphate-buffered saline (PBS). We have clarified this in the revised Methods section.

Location of the changes: Page 13, Lines 246-248.

Comment #3: At line 301, please specify the manufacturer of Abmart.

Author's reply: We have now specified the manufacturer for the anti-LC3 antibody from Abmart in the revised text.

Location of the changes: Page 15, Lines 305.

Comment #4: For Figure 3C, it would be helpful to include representative images demonstrating staining intensity scores (0–3+). Please also clarify whether the graph represents a combined score and, if possible, provide a breakdown of staining intensity across normal and tumour tissues. In addition, please confirm that all images are at 100× magnification, as there appear to be differences in cell size across panels.

Author's reply: We sincerely thank the reviewer for the insightful suggestions, which have helped us improve the clarity of our manuscript. We have addressed each point as follows.

1. Clarification of the scoring method and confirmation of magnification:

As detailed in the Methods section, we used a semi-quantitative scoring system evaluating both intensity (0-3) and extent (0-5), with a final combined score ranging from 0 to 8. To prevent any ambiguity, we have now explicitly stated in the revised legend of Figure 3C: "Data are presented as the combined immunohistochemistry score (staining intensity score + extent score)." We confirm that all representative photomicrographs in Figure 3C were acquired at a consistent 100× original magnification.

Location of the changes: Page 20, Lines 410-413.

2. Regarding the request for a breakdown of staining intensity:

We appreciate the reviewer's suggestion to provide a more granular view of the intensity distribution. Our semi-quantitative analysis was designed to yield the validated combined score (0-8) as the primary metric for statistical comparison, as described. The scoring was performed directly to assign this final composite value per sample. We acknowledge that presenting the separate distribution of the four intensity sub-categories (0, 1+, 2+, 3+) would offer an additional perspective. However, to ensure scoring consistency and minimize intra-/inter-observer variability, the original data were recorded and analyzed as the integrated final score. A retrospective, accurate re-analysis to disaggregate the contribution of each intensity level across all samples is not feasible without introducing potential bias and is beyond the scope of the current revision.

3. Validation of the presented data:

We assure the reviewers that the data provided strongly supports our conclusion. The comparison of staining intensities between the "Normal" and "Tumor" panels in Figure 3C can visually confirm the significant increase in the comprehensive score of the tumor tissue. Tumor images uniformly show stronger and more extensive staining, corresponding to higher intensity (2+ or 3+) and range scores.

We believe that the current method and presentation employ a standard and transparent scoring system, providing clear and conclusive evidence for differential protein expression. Thank you to the reviewers for your valuable suggestions. We will take these suggestions into account in the design of future research.

Comment #5: In the Results section, when presenting immunohistochemistry data, please use caution when describing “upregulated expression” and clearly explain how the quantitative IHC results were derived.

Author's reply: We agree with the reviewer and have revised the description to be more precise. The revised text now avoids the term “upregulated” and explicitly states that the comparison is based on quantitative H-score analysis of paired samples. The updated text in the Results section reads: “To evaluate RUBCN expression at the protein level, we performed immunohistochemical (IHC) analysis on five paired samples of primary breast cancer and adjacent normal tissues. Quantitative IHC scoring revealed that RUBCN protein levels were significantly higher in tumor tissues compared with their paired normal counterparts (Figure 3C).”

This revision provides a clearer and more accurate description of the IHC data and its quantification.

Location of the changes: Page 20, Lines 401-405.

Comment #6: Please include the RUBCN antibody validation using the knockout control in Figure 9E to further support the specificity and robustness of these findings.

Author's reply: This is an excellent suggestion. We have now performed the recommended control experiment. Specifically, we conducted an additional Western blot using the same protein lysates from RUBCN-knockdown MDA-MB-231 cells to verify the knockdown efficiency. The updated Figure 9E now includes this RUBCN blot, which confirms the successful knockdown. The incorporation of this validation data strengthens the specificity of our experimental approach and the robustness of the corresponding conclusions. We sincerely appreciate the reviewer's insightful comment.

---

## [Decision Letter · Decision Letter 2]

6 Jan 2026

RUBCN as a Novel Prognostic Biomarker and Therapeutic Target in Breast Cancer

PONE-D-25-46002R2

Dear Dr. Huang,

We’re pleased to inform you that your manuscript has been judged scientifically suitable for publication and will be formally accepted for publication once it meets all outstanding technical requirements.

Kind regards,

Amy McCart Reed

Academic Editor

PLOS One

Additional Editor Comments (optional):

Reviewers' comments:

Reviewer's Responses to Questions

**Comments to the Author**

Reviewer #2: All comments have been addressed

2. Is the manuscript technically sound, and do the data support the conclusions?

Reviewer #2: Yes

3. Has the statistical analysis been performed appropriately and rigorously?

Reviewer #2: Yes

4. Have the authors made all data underlying the findings in their manuscript fully available?

Reviewer #2: Yes

5. Is the manuscript presented in an intelligible fashion and written in standard English?

Reviewer #2: Yes

Reviewer #2: (No Response)

**Do you want your identity to be public for this peer review?** For information about this choice, including consent withdrawal, please see our Privacy Policy

Reviewer #2: No

---

## [Editor Report · Acceptance letter]

PONE-D-25-46002R2

PLOS One

Dear Dr. Huang,

I'm pleased to inform you that your manuscript has been deemed suitable for publication in PLOS One. Congratulations! Your manuscript is now being handed over to our production team.

Kind regards,

on behalf of

Associate Professor Amy McCart Reed

Academic Editor

PLOS One